# Towards Optimism-Pessimism Trade-off in Model-based Offline-to-Online Reinforcement Learning

## Abstract

Model-based offline-to-online reinforcement learning (RL) provides a sample-efficient framework by pre-training environment models and control policies using offline data, followed by fine-tuning through limited online interactions. However, the distribution shifts between offline and online stages often hinders fine-tuning performance. Existing methods approach this problem by adjusting the trade-off between optimism and pessimism using a single-objective formulation, which requires online evaluation across tasks. This results in an expensive bi-level optimization procedure. In this work, we identify this optimism-pessimism trade-off during offline training as a key challenge: optimistic policies tend to generalize better to novel online tasks by exploring out-of-distribution states and actions, while pessimistic policies remain constrained to the offline data distribution and perform better on tasks that are similar to the offline tasks. To address this challenge, we propose a bi-objective formulation that captures this trade-off and yields a pool of Pareto policies during offline training. These policies reflect varying levels of trade-offs, enabling flexible selection of policies for various online tasks. To produce these policies, we introduce Multiple-Objective Soft Actor-critIC (MOSAIC), which solves multiple bi-objective optimization problems guided by reference vectors and refines the Pareto policy pool through neighborhood search. After offline training, a contextual bandit algorithm hierarchically selects the most suitable policy for fine-tuning at each online interaction step. Empirically, our pipeline, **Hi**erarchical **P**areto **P**olicy **P**ool (**HiP3**), achieves state-of-the-art performance on offline-to-online RL benchmarks with diverse online tasks. Comprehensive ablation studies are conducted to further elucidate the mechanisms behind HiP3.

## 1 Introduction

A critical barrier of applying reinforcement learning (RL) in various real-world scenarios is its high sample complexity (Yu et al., 2018; Gottesman et al., 2019; Schulman et al., 2015). Recent studies have focused on learning policies only using pre-collected data. However, directly deploying these offline-trained policies to online environments often results in suboptimal performance due to the distribution shift between the pre-collected data and data collected through online interactions, leading to the emergence of offline-to-online RL (O2O RL) (Nakamoto et al., 2024; Zhang et al., 2023). Among existing methods, model-based O2O RL approaches demonstrate better sample efficiency compared to their model-free counterparts, since they first learn environment models and control policies from offline data and subsequently fine-tune them through minimal online interactions.

Despite promising results (e.g., (Nakamoto et al., 2024; Zhang et al., 2023)), the distribution shift between offline and online stages still hinders the fine-tuning performance of model-based O2O RL (Mao et al., 2022; Rafailov et al., 2023). In this paper, we identify the optimism-pessimism trade-off during the offline stage as a primary challenge, illustrated through a proof-of-concept experiment in Figure 1. Following prior works (Yu et al., 2020; Kidambi et al., 2020; Yang et al., 2021), we utilize the model-predicted return and its uncertainty estimation as proxies of optimism and pessimism objectives, respectively. During the offline stage, a pool of Pareto policies is generated to achieve diverse trade-offs between these two objectives. As shown in Figure 1 (b), when fine-tuning these policies on novel online tasks, the optimistic Pareto policy 1 (higher model return) exhibits superior performance as it is better optimized during the offline stage to explore states and actions

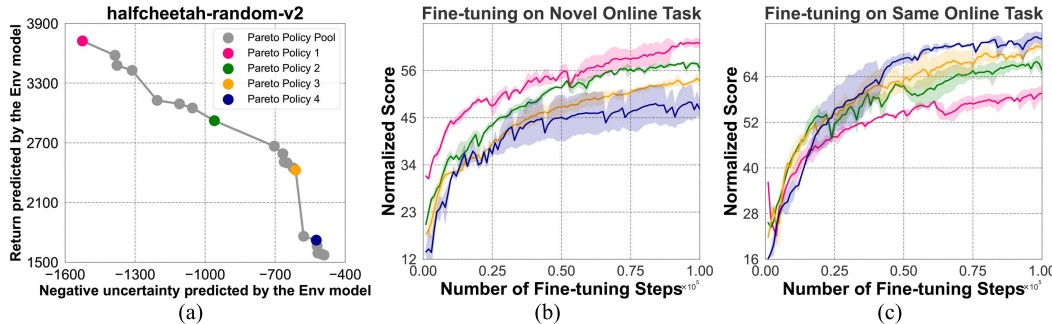

Figure 1: (a) A Pareto policy pool performing diverse optimism-pessimism trade-offs achieved by our HiP3 on the halfcheetah-random-v2 task from the D4RL Gym benchmark Fu et al. (2020). (b) The fine-tuning curves of diverse policies on the novel online task (a task altered from halfcheetah). Optimistic Pareto policy 1 with higher model return achieves higher fine-tuning performance. (c) The fine-tuning curves of diverse policies on the same online task of halfcheetah. Pessimistic Pareto policy 4 with lower model uncertainty performs better.

beyond the distribution of pre-training data. Conversely, as shown in Figure 1(c), the pessimistic Pareto policy 4 (lower model uncertainty) remains constrained to the distribution of pre-training data, and hence performs adequately only on the same or similar online tasks. Pareto policies 2&3 demonstrates a more balanced optimism-pessimism trade-off, performing competitively across both types of tasks. These observations indicate the importance of the optimism-pessimism trade-off to the performance of model-based O2O RL. However, determining the optimal trade-off poses a significant challenge for existing methods, which often adjust this trade-off within a single combined objective (e.g., by tuning a trade-off hyperparameter) and assess post-tuning performance across various tasks through online RL, resulting in a costly bi-level optimization problem.

In this paper, we instead treat optimism and pessimism as two distinct objectives and propose a bi-objective formulation for the trade-off, which provably yields a pool of diverse policies covering the entire Pareto front during the offline stage. These policies exhibit varying trade-off levels, thereby providing the flexibility to select the most suitable policy for fine-tuning on diverse online tasks. Several algorithmic challenges arise in generating this Pareto policy pool and in selecting the appropriate policies from the pool: (1) How to find policies performing diverse optimism-pessimism trade-offs across various Pareto front regions? (2) Given this Pareto policy pool, how can we dynamically select the most suitable policy for each state during online fine-tuning? To address challenge (1), we introduce a Multi-Objective Soft Actor-critIC (MOSAIC) algorithm, which generates a diverse Pareto policy pool by first solving multiple bi-objective policy optimization problems with varying constraints defined by the reference vectors targeting diverse regions of the Pareto front, and then employing a neighborhood search method to identify additional Pareto-optimal policies around the obtained ones. To address challenge (2), a contextual bandit algorithm is adapted to hierarchically select a suitable policy from the pool at each interaction step for online fine-tuning.

We evaluate our method, Hierarchical Pareto Policy Pool (HiP3), against several state-of-the-art online and offline-to-online RL methods using the standard D4RL Gym benchmark (Fu et al., 2020) with diverse online tasks. HiP3 achieves the highest average score across multiple datasets and tasks, and outperforms other baseline methods when fine-tuned on novel online tasks, illustrating its efficacy in addressing the optimism-pessimism trade-off. Additionally, a comprehensive ablation study is conducted to determine the critical components contributing to the performance of HiP3.

## 2 RELATED WORK

**Offline RL** Traditional RL algorithms optimize policies through direct interactions with the environment. However, in complex environments, such approaches often necessitate a substantial number of interactions to learn an effective policy, thereby imposing considerable computation overhead. To address this challenge, researchers have developed offline RL, which allows the optimization of policies using pre-collected data, thereby alleviating the need for online interactions. However, directly optimizing policies with pre-collected data may result in suboptimal performance because these policies might be learnt from out-of-distribution (OOD) state-action pairs, leading to inaccurate value estimations. Current approaches address this issue through various techniques, including explicit (Fujimoto et al., 2019; Jaques et al., 2019; Wu et al., 2019) and implicit (Peng et al., 2019; Siegel et al., 2020) policy constraints, uncertainty estimation (Wu et al., 2019; Kumar et al., 2019),

imitation learning (Siegel et al., 2020), and regularization (Wang et al., 2020; Kumar et al., 2020). These approaches rely on the idea of *pessimism*, ensuring that the learned policy avoids uncertain regions that are not well covered by the offline data.

**Offline-to-Online RL** Due to the distribution shift between offline and online environments (Koh et al., 2021), directly deploying an offline-trained policy to an online environment may result in poor performance. Therefore, fine-tuning these policies online is necessary for performance improvement. However, directly fine-tuning these policies may still incur unsatisfactory performance (Nair et al., 2020) because (1) the distribution shift between offline and online environments can destabilize policy fine-tuning (Lee et al., 2022), and (2) the inherent pessimism of offline RL constrains its capacity to explore unknown environments when transitioned from offline to online. For challenge (1), existing algorithms utilize balanced replay (Lee et al., 2022), priority sampling (Zheng et al., 2023), and policy expansion (Zhang et al., 2023) to address the training instability issue. For challenge (2), (Mark et al., 2022) suggests employing a critic function to guide exploration. Nevertheless, this method often attempts to balance the optimism and pessimism trade-off in a single scenario, which may be inadequate for fine-tuning across diverse online tasks. HiP3 overcomes this limitation by offering a diverse set of policies with varying optimism-pessimism trade-offs to suit different tasks and develops a hierarchical RL framework with a contextual bandit for state-dependent policy selection during online fine-tuning.

**Model-based RL** Model-based RL improves sample efficiency by constructing an environment model, thereby minimizing the need to interact with real environments. For tasks with low dimensionality, linear models and Gaussian processes are often sufficient to model the environment dynamics. For high-dimensional tasks, deep neural networks are more appealing because they are high-capacity function approximators that can leverage large-scale datasets more effectively for environment modeling. However, directly optimizing policies based on learned environment models is susceptible to model exploitation (Janner et al., 2019; Lu et al., 2021). Previous efforts have focused on enhancing robustness (Lee et al., 2021) and adapting to distribution shifts (Hishinuma and Senda, 2021) to refine the dynamics of policy learning. Recent approaches restrict policies to approximate behavioral policies by employing environment models for planning (Argenson and Dulac-Arnold, 2020) and policy optimization (Matsushima et al., 2020; Yu et al., 2021). Furthermore, uncertainty quantification techniques have been integrated into environment models to facilitate learning policies beyond the scope of data coverage (Yu et al., 2020; Kidambi et al., 2020).

## 3 PRELIMINARIES

In standard RL, the interaction between an agent and an environment is formulated as a Markov Decision Process (MDP) (Sutton and Barto, 2018). An MDP is defined by a tuple $(S, A, p, r, \rho_0, \gamma)$, where $S$ denotes the state space, $A$ denotes the action space, $p(s'|s, a)$ corresponds to the state transition probability, $r(s, a)$ constitutes the reward function, $\rho_0$ specifies the initial state distribution, and $\gamma \in [0, 1)$ is the discount factor. The goal of reinforcement learning is to find a policy $\pi$ that maximizes the expected sum of discounted rewards:

$$J(\pi_\theta) = \mathbb{E}_{\rho_0, \pi_{\theta \in \Theta}} \left[ \sum_{t=0}^{+\infty} \gamma^t r(s_t, a_t) \right], \tag{1}$$

where $\pi_\theta$ denotes the policy, parameterized by $\theta \in \Theta$, that given state $s_t$ yields action $a_t$ at time $t$.

Specifically, model-based O2O RL methods first trains environment models and control policies using offline pre-collected data, and then fine-tunes them with the limited online interaction data collected from real environment (Mao et al., 2022; Rafailov et al., 2023). The environment model, typically parameterized by a feed-forward neural network, contains a learned transition probability model $\hat{p}(s'|s, a)$ and a reward model $\hat{r}(s, a)$, which are used to predict the next state $s' \sim \hat{p}(s'|s, a)$ and reward value $\hat{r}$ based on the current state $s$ and action $a$ selected by policy $\pi_\theta(s)$.

As discussed above and shown in Figure 1, the optimism-pessimism trade-off plays a vital role in determining the performance of model-based O2O RL. Following prior works (Yu et al., 2020; Kidambi et al., 2020; Yang et al., 2021), we employ the model-predicted reward $\hat{r}(s, a)$ and its uncertainty estimation $u(s, a)$ as surrogate objectives for optimism and pessimism, respectively [1]. Existing methods usually tune the trade-off between these two objectives in a single objective, i.e.,

---

[1] Further information on the uncertainty estimation $u$ is provided in Appendix A.4, and a description of the environment model architecture is given in Appendix A.5.

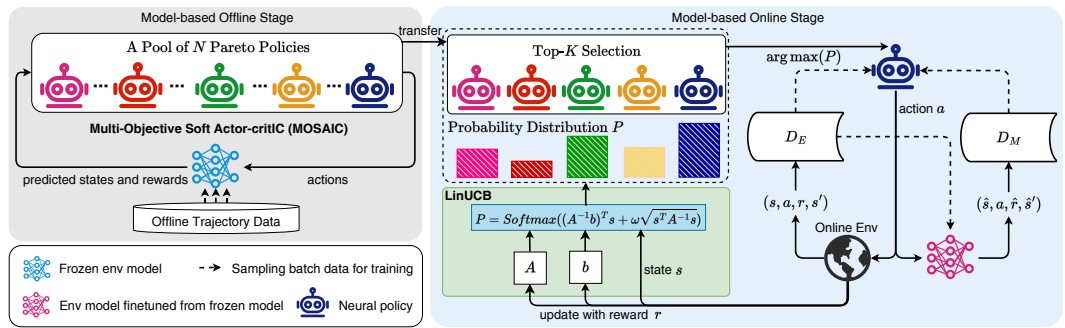

Figure 2: **Overview of Hierarchical Pareto Policy Pool (HiP3)**. HiP3 trains a pool of diverse Pareto policies using our MOSAIC algorithm, performing various levels of optimism-pessimism trade-offs (Grey box). A contextual bandit algorithm (i.e., LinUCB (Chu et al., 2011)) is then used to select the best policy from the Top-$K$ policies in the pool for fine-tuning at each online interaction step (Blue box). The selected policy is then optimized using a Dyna-style RL method (Janner et al., 2019). The contextual bandit proceeds to select the next policy for online fine-tuning.

$\bar{r}(s,a) = \hat{r}(s,a) - \lambda u(s,a)$, and evaluate post-tuning performance on different online tasks using model-based online RL, leading to an expensive bi-level policy optimization problem, where the outer loop needs to run many instances of model-based online RL (inner loop).

Instead of solving this expensive bi-level optimization problem (See Appendix A.7 for detailed analysis of using HiP3 rather than Linear Scalarization method), we treat $\hat{r}$ and $u$ as two distinct objectives and solve a bi-objective optimization problem as following:

$$\max_{\theta} J(\pi_{\theta}) = \max_{\theta} \left( J^{\hat{r}}(\pi_{\theta}), J^{\hat{u}}(\pi_{\theta}) \right) \triangleq \mathbb{E}_{\rho_0, \pi_{\theta}} \left[ \sum_{t=0}^{+\infty} \gamma^t \left( \hat{r}(s_t, a_t), \hat{u}(s_t, a_t) \right) \right], \qquad (2)$$

where $\hat{u}(s_t, a_t) \triangleq \exp(-u(s_t, a_t)/\kappa)$ and $\kappa$ is a temperature coefficient that smooths the loss landscape. This formulation eliminates the need of manually tuning a trade-off coefficient (e.g., $\lambda$ in $\bar{r}$), and enables the generation of a diverse set of policies that lie along the Pareto front (See Appendix A.6 for a brief introduction of multi-objective optimization.). Each policy inherently reflects a different optimism-pessimism trade-off, offering the flexibility to adaptively select suitable policies to fine-tune for various online tasks. However, efficiently solving Eq. 2 and selecting appropriate policies for online adaptation remains challenging. To this end, we propose HiP3, a hierarchical Pareto policy pool method, that incorporates two algorithmic innovations for model-based O2O RL.

## 4    HIP3: HIERARCHICAL PARETO POLICY POOL

Figure 2 illustrates the main idea of HiP3 alongside the pseudocode provided in Algorithm 1. To enhance the fine-tuning performance across various online tasks, HiP3 constructs a Pareto policy pool during the offline RL stage, containing policies exhibiting various degrees of the optimism-pessimism trade-offs. To realize these policies, we implement a Multi-Objective Soft Actor-CritIC (MOSAIC) algorithm that extends Soft Actor-Critic (SAC) (Haarnoja et al., 2018a) for multi-objective optimization (Lin et al., 2019; Mahapatra and Rajan, 2020; Chen and Kwok, 2022). This algorithm solves several bi-objective policy optimization problems with the constraints uniquely determined by the reference vectors targeting distinct regions of the Pareto front, and subsequently a neighborhood search method is employed to discover additional Pareto-optimal policies around the achieved ones (Grey box in Figure 2). After the offline RL stage, HiP3 utilizes a contextual bandit algorithm to hierarchically select a suitable policy from the pool for fine-tuning at each online interaction step (Blue box in Figure 2).

### 4.1    MOSAIC: GENERATION OF PARETO POLICY POOL

The bi-objective policy optimization problem, as described in Eq. 2, theoretically contains an infinite number of Pareto-optimal policies, making it infeasible in practice to find the entire Pareto set. Inspired by prior works (Lin et al., 2019; Mahapatra and Rajan, 2020; Chen and Kwok, 2022), we explore the Pareto front by generating a diverse set of reference vectors that define several constrained bi-objective optimization problems. Specifically, we produce $n$ uniformly distributed reference vectors $\{v_i\}_{i=1}^{n}$ within a 0-1 normalized objective space:

**Algorithm 1** Hierarchical Pareto Policy Pool (HiP3)

1: **Input:** Offline dataset $D$, #reference vectors $n$, #offline policies $K_{\text{off}}$, #online policies $K_{\text{on}}$, #offline steps $T_{\text{off}} = n(T_g + 2T_l)T_e$, #online steps $T_{\text{on}}$, model update interval $S_M$, bandit hyperpara. $\omega$

2: **Initialize:** Pareto policy pool $\mathcal{P} = \emptyset$, env. model $M$, data buffer from model $D_M = \emptyset$, data buffer from env. $D_E = \emptyset$

// Offline learning via MOSAIC

3: Train $M$ on dataset $D$ (max. likelihood)

4: Generate reference vectors $\{\boldsymbol{v}_1, \ldots, \boldsymbol{v}_n\}$ (Eq. 3)

5: **for** $i = 1$ to $n$ **do** //with $D_M$ accumulated

6:     Initialize a policy $\pi_\theta$

7:     Update $\pi_\theta$ for $T_g \times T_e$ steps with $\boldsymbol{v}_i$ (Alg. 2)

8:     $\boldsymbol{v}_i^+ \leftarrow \boldsymbol{v}_i + \epsilon$; $\boldsymbol{v}_i^- \leftarrow \boldsymbol{v}_i - \epsilon$

9:     **for** $\boldsymbol{v}' \in \{\boldsymbol{v}_i^+, \boldsymbol{v}_i^-\}$ **do**

10:       **for** $j = 1$ to $T_l$ **do**

11:         Update $\pi_\theta$ for $T_e$ steps with $v'$ (Alg. 2)

12:         $\mathcal{P} \leftarrow \mathcal{P} \cup \{\pi_\theta\}$

13:         $\mathcal{P} \leftarrow$ top-$K_{\text{off}}(\mathcal{P})$ //via non-dominant sort

// Online selection via LinUCB (Chu et al., 2011)

14: $D_M = \emptyset$

15: Select top-$K_{\text{on}}$ policies from $\mathcal{P}$ as $\mathcal{P}_{on}$ via eval.

16: Construct contextual bandit $B(A, b)$ with $\mathcal{P}_{on}$

17: **for** $t = 1$ to $T_{\text{on}}$ **do**

18:     **if** $t \bmod S_M == 0$ **then**

19:       Train $M$ on $D_E$ (max. likelihood)

20:       Obtain $\tau_M$ via $k$-step rollout on $M$ using $B$ and $\mathcal{P}_{on}$

21:       $D_M \leftarrow D_M \cup \{\tau_M\}$

22:     $P \leftarrow \text{Softmax}((A^{-1}b)^T s_t + \omega \sqrt{s_t^T A^{-1} s_t})$

23:     $\pi'_\theta \leftarrow \arg\max(P)$

24:     Execute $\pi'_\theta$ in env. $E$ and observe reward $r$ and transition $\tau$

25:     $A \leftarrow A + s_t s_t^T$; $b \leftarrow b + s_t r_t$

26:     $D_E \leftarrow D_E \cup \{\tau\}$

27:     Use SAC (Haarnoja et al., 2018a) to update $\pi'_\theta$ with batch samples from $D_E \cup D_M$

$$\boldsymbol{v}_i = (v_i^1, v_i^2) = (\tau_b - (i-1)\tau_c, \tau_a + (i-1)\tau_c)$$

$$\text{with } \tau_c = \frac{\tau_b - \tau_a}{n-1}, 0 < \tau_a < \tau_b < 1, i \in \{1, \cdots, n\},$$

where $\tau_a$ and $\tau_b$ control the range covered by the reference vectors. Each $\boldsymbol{v}_i = (v_i^1, v_i^2)$ defines a constrained bi-objective optimization problem, as described below, whose solution located within the target region of the Pareto front:

$$\max_\theta \boldsymbol{J}(\pi_\theta) = \max_\theta \left( J^{\hat{r}}(\pi_\theta), J^{\hat{u}}(\pi_\theta) \right)$$

$$\text{s.t. } \Psi(\pi_\theta, \boldsymbol{v}_i) = D_{\cos}\left( \frac{\boldsymbol{J}(\pi_\theta)}{\|\boldsymbol{J}(\pi_\theta)\|_2}, \frac{\boldsymbol{v}_i}{\|\boldsymbol{v}_i\|_2} \right) \geq \psi,$$

where $\Psi$ defines the cosine similarity between the objective $\boldsymbol{J}(\pi_\theta)$ and the reference vector $\boldsymbol{v}_i$, and the hyper-parameter $\psi$ controls the scope of the target region. To solve this bi-objective optimization problem with an inequality constraint, we introduce MOSAIC, a Multi-Objective Soft Actor-Critic algorithm, with the pseudocode provided in Algorithm 2. As an off-policy algorithm, MOSAIC attains the Pareto policy by simultaneously maximizing $Q$ values of $\hat{r}$ and $\hat{u}$,

$$Q_{\phi^{\hat{r}}}^{\hat{r}}(s_t, a_t) = \hat{r}(s_t, a_t) + \gamma \mathbb{E}\left[ Q_{\phi^{\hat{r}}}^{\hat{r}}(s_{t+1}, a_{t+1}) \right],$$

$$Q_{\phi^{\hat{u}}}^{\hat{u}}(s_t, a_t) = \hat{u}(s_t, a_t) + \gamma \mathbb{E}\left[ Q_{\phi^{\hat{u}}}^{\hat{u}}(s_{t+1}, a_{t+1}) \right]. \tag{3}$$

During each update iteration, these $Q$ functions along with their corresponding target $\overline{Q}$ functions are updated using batches sampled from the model buffer $D_M$, as outlined in Lines 8-11 of Algorithm 2. Subsequently, MOSAIC evaluates the degree of constraint violation (Line 12) and selectively applies gradient ascent to the objective in order to satisfy the constraint (*Correction stage*). Upon satisfying the constraint, the algorithm transitions to the *Ascent stage*, thereby reinterpreting the constrained bi-objective problem as a tri-objective problem by setting the constraint as the third objective:

$$\max_\theta \boldsymbol{F}(\pi_\theta) \triangleq \max_\theta \left( J^{\hat{r}}(\pi_\theta), J^{\hat{u}}(\pi_\theta), \Psi(\pi_\theta, \boldsymbol{v}_i) \right). \tag{4}$$

To solve this tri-objective optimization problem, we employ the MGDA algorithm (Désidéri, 2012), which devises a convex combination of the gradients of all three objectives at each update step, ensuring that the resulting update direction $\boldsymbol{\beta}_t \nabla_{\theta_t} \boldsymbol{F}(\pi_{\theta_t})$, where $\boldsymbol{\beta}_t = [\beta_t^{\hat{r}}, \beta_t^{\hat{u}}, \beta_t^\psi]$, does not decrease any of the three objectives. This process is equivalent to solving the following min-norm problem:

$$\min_{\boldsymbol{\beta}_t} \|\boldsymbol{\beta}_t \nabla_{\theta_t} \boldsymbol{F}(\pi_{\theta_t})\|_2 \quad s.t. \quad \|\boldsymbol{\beta}_t\|_1 = 1, \boldsymbol{\beta}_t \geq 0, \tag{5}$$

which can be effectively tackled via the Frank-Wolfe algorithm (Jaggi, 2013). Subsequently, we perform the policy gradient update as specified in Line 19 of Algorithm 2. Given that MGDA systematically enhances all objectives, it guarantees that the constraints remain unviolated, thereby ensuring that the algorithm eventually converges to a Pareto-optimal policy within the targeted

---

**Algorithm 2** Multi-Objective Soft Actor-critIC (MOSAIC)

1: **Input:** reference vector $v$, constraint $\psi$, #iterations $T$, env. model $M$, data buffer $D_M$ and its update interval $S_M$

2: **Initialize:** $\phi_i^{\hat{r}}$, $\phi_i^{\hat{u}}$, $\overline{\phi_i^{\hat{r}}}$, $\overline{\phi_i^{\hat{u}}}$ for $i \in \{1, 2\}$

3: **for** $t = 1$ to $T$ **do**

4:     **if** $t \bmod S_M == 0$ **then**

5:         Obtain $\tau_M$ with $k$-step rollout on $M$ by $\pi_\theta$

6:         $D_M \leftarrow D_M \cup \{\tau_M\}$

7:     Sample a batch from $D_M$

8:     $\phi_i^{\hat{r}} \leftarrow \phi_i^{\hat{r}} - \eta \nabla_{\phi_i^{\hat{r}}} J^{Q^{\hat{r}}}(\phi_i^{\hat{r}})$ for $i \in \{1, 2\}$

9:     $\phi_i^{\hat{u}} \leftarrow \phi_i^{\hat{u}} - \eta \nabla_{\phi_i^{\hat{u}}} J^{Q^{\hat{u}}}(\phi_i^{\hat{u}})$ for $i \in \{1, 2\}$

10:     $\overline{\phi_i^{\hat{r}}} \leftarrow \tau \phi_i^{\hat{r}} + (1 - \tau)\overline{\phi_i^{\hat{r}}}$ for $i \in \{1, 2\}$

11:     $\overline{\phi_i^{\hat{u}}} \leftarrow \tau \phi_i^{\hat{u}} + (1 - \tau)\overline{\phi_i^{\hat{u}}}$ for $i \in \{1, 2\}$

// Correction stage

12:     **if** $\Psi(\pi_\theta, \boldsymbol{v}) < \psi$ **then**

13:         **if** $\frac{J^{\hat{r}}(\pi_\theta)}{J^{\hat{u}}(\pi_\theta)} < \frac{v^1}{v^2}$ **then**

14:             $\pi_\theta \leftarrow \pi_\theta + \eta \nabla_\theta J^{\hat{r}}(\pi_\theta)$

15:         **else**

16:             $\pi_\theta \leftarrow \pi_\theta + \eta \nabla_\theta J^{\hat{u}}(\pi_\theta)$

// Ascent stage

17:     **else**

18:         Find $\boldsymbol{\beta}^*$ according to Eq. 5

19:         $\pi_\theta \leftarrow \pi_\theta + \eta \boldsymbol{\beta}^* \nabla_\theta \boldsymbol{F}(\pi_\theta)$

20:     $\alpha \leftarrow \alpha - \lambda \nabla_\alpha J(\alpha)$ //SAC temperature $\alpha$ update

21: **Output:** policy $\pi_\theta$

---

region. A complete proof is provided in Appendix A.3, where Theorem 1 establishes the convergence rate of the ascending stage of Algorithm 2. For the correction stage, which follows the standard single-objective SAC update, the convergence result has already been established in (Haarnoja et al., 2018b).

As discussed above, MOSAIC is capable of generating a variety of Pareto-optimal policies by employing diverse reference vectors. However, the computational cost grows linearly with an increased number of reference vectors that are required to govern the updates for more policies. Practically, a limited set of Pareto policies fails to encompass the full spectrum of trade-offs, potentially resulting in sub-optimal policy selection in the online stage. To address this problem, we perturb the reference vector $\boldsymbol{v}_i$ to create augmented reference vectors $\boldsymbol{v}_i^+$ and $\boldsymbol{v}_i^-$ in Line 8 of Algorithm 1, thereby constructing additional constrained bi-objective optimization problems based on Eq. 3. By solving these bi-objective problems in the neighborhood of the original ones, more Pareto-optimal policies are discovered. This neighborhood search method leads to a broader range of Pareto policies that exhibit diverse optimism-pessimism trade-offs, while maintaining an acceptable computational cost.

## 4.2 Online Policy Selection and Optimization

Given a set of Pareto policies produced in the offline stage, determining the optimal policy for various online tasks necessitates a strategic approach. A straightforward method involves evaluating each policy through online RL and selecting the one policy that attains the highest return. However, this approach requires extensive interaction with the environment, particularly when the pool of policies to be assessed is extensive.

A more effective alternative involves developing a high-level policy that selects the most suitable one from the pool of low-level policies for fine-tuning at each online interaction step. To this end, we propose a hierarchical RL method that can adopt any conventional contextual bandit algorithm as the high-level policy (refer to Lines 14-27 of Algorithm 1 and the blue box in Figure 2). In order to enhance fine-tuning performance and reduce computation cost, we select the Top-$K$ policies from the Pareto policy pool based on how they perform on the chosen online task.The performance curves in Figure 4 show that LinUCB (Chu et al., 2011) achieves superior performance compared to all other candidate bandit algorithms.

Specifically, at each online interaction step $t$, LinUCB calculates the probability distribution over the selected Top-$K$ policies based on the current state $s_t$ and selects the one with the highest likelihood (Lines 22-23 of Algorithm 1). The chosen policy then interacts with the online environment to receive the corresponding reward $r_t$, which is subsequently used to update the parameters $(A, b)$ of LinUCB (Line 25 of Algorithm 1). More details about our bandit algorithm are provided in Appendix A.8.

It is worth noting that, unlike previous O2O RL methods (Nair et al., 2020; Zhang et al., 2023; Nakamoto et al., 2024), which rely on retaining the offline dataset $D$ for online fine-tuning (i.e., sampling data batches from both the offline dataset $D$ and the online buffer $D_E$, or initializing the online buffer directly with $D$), our approach eliminates the need for offline dataset $D$ during the online RL stage (Line 19 and Line 27 of Algorithm 1) and enables rapid adaptation to the online environment, while achieving superior performance compared to prior methods.

| | | On the Same Online Tasks | | | | | On Novel Online Tasks | | | | |
|---|---|---|---|---|---|---|---|---|---|---|---|
| | | AWAC | PEX | CAL-QL | MBPO | HiP3 (Ours) | AWAC | PEX | CAL-QL | MBPO | HiP3 (Ours) |
| Random | HalfCheetah | 35.6 ±0.5 | 56.9 ±1.2 | 15.4 ±5.6 | 70.3 ±12.7 | 82.8 ±1.1 | 33.2 ±0.4 | 42.2 ±2.6 | 11.1 ±2.5 | 50.0 ±21.3 | 63.8 ±1.8 |
| | Hopper | 11.7 ±0.2 | 22.2 ±4.1 | 12.7 ±8.5 | 98.9 ±6.0 | 107.7 ±1.6 | 9.2 ±0.1 | 23.5 ±17.2 | 11.7 ±4.7 | 85.1 ±3.8 | 89.7 ±2.7 |
| | Walker2d | 12.2 ±6.7 | 9.8 ±2.2 | 1.6 ±1.3 | 52.4 ±26.9 | 30.2 ±25.6 | 7.0 ±1.3 | 8.5 ±0.7 | -0.6 ±1.1 | 44.1 ±11.4 | 41.3 ±27.8 |
| Medium | HalfCheetah | 43.6 ±0.8 | 44.1 ±0.3 | 49.7 ±0.3 | 70.3 ±12.7 | 89.1 ±5.1 | 33.9 ±0.4 | 30.4 ±0.9 | 35.4 ±0.1 | 50.0 ±21.4 | 69.9 ±1.8 |
| | Hopper | 101.5 ±0.4 | 99.4 ±1.2 | 83.3 ±12.3 | 98.9 ±6.0 | 107.7 ±1.6 | 88.5 ±0.1 | 74.8 ±15.9 | 37.6 ±4.6 | 85.1 ±3.8 | 92.8 ±1.6 |
| | Walker2d | 79.0 ±1.7 | 72.0 ±1.2 | 83.1 ±2.2 | 52.4 ±26.9 | 70.4 ±17.5 | 70.4 ±0.3 | 61.5 ±1.0 | 59.6 ±0.4 | 44.1 ±11.4 | 62.8 ±5.8 |
| Medium-replay | HalfCheetah | 45.3 ±0.1 | 45.9 ±0.1 | 47.8 ±0.1 | 70.3 ±12.7 | 92.5 ±0.9 | 29.9 ±3.8 | 30.0 ±3.5 | 32.6 ±0.2 | 50.0 ±21.3 | 72.0 ±1.2 |
| | Hopper | 33.9 ±1.4 | 36.0 ±1.4 | 95.1 ±1.4 | 98.9 ±6.0 | 102.9 ±2.1 | 29.6 ±2.8 | 39.4 ±14.7 | 65.9 ±14.0 | 85.1 ±3.8 | 92.3 ±3.2 |
| | Walker2d | 34.5 ±8.4 | 40.6 ±7.3 | 85.5 ±4.9 | 52.4 ±26.9 | 96.4 ±12.5 | 20.0 ±3.2 | 38.0 ±4.3 | 46.4 ±3.7 | 44.1 ±11.4 | 81.1 ±8.1 |
| Medium-expert | HalfCheetah | 103.9 ±1.5 | 48.3 ±5.6 | 92.5 ±1.6 | 70.3 ±12.7 | 91.1 ±3.4 | 77.6 ±0.7 | 29.9 ±1.0 | 64.4 ±1.9 | 50.0 ±21.3 | 74.8 ±2.6 |
| | Hopper | 112.3 ±0.4 | 89.0 ±31.6 | 112.2 ±0.2 | 98.9 ±6.0 | 105.7 ±5.7 | 88.8 ±4.4 | 50.9 ±9.5 | 88.0 ±5.5 | 85.1 ±3.8 | 93.6 ±0.8 |
| | Walker2d | 105.8 ±1.8 | 108.1 ±0.5 | 109.5 ±0.7 | 52.4 ±26.9 | 76.5 ±12.8 | 50.6 ±8.5 | 54.3 ±2.8 | 79.2 ±3.4 | 44.1 ±11.4 | 75.0 ±3.5 |
| Total Mean | | 60.0 ±2.0 | 53.5 ±4.7 | 65.7 ±3.3 | 73.9 ±15.2 | 87.7 ±8.4 | 48.0 ±4.9 | 40.3 ±6.2 | 44.3 ±3.6 | 59.7 ±12.2 | 75.8 ±5.1 |

Table 1: Performance comparison on the same and novel online tasks. The novel online tasks are generated by adjusting gravity, friction, noisy scales, and reward functions of the MuJoCo environments. We fine-tune all the methods with the same hyper-parameter settings provided by the original papers. Scores for each dataset are normalized according to the D4RL benchmark (Fu et al., 2020): **Normalized Score** $= 100 \times \frac{\text{return} - \text{random return}}{\text{expert return} - \text{random return}}$. Full results can be found in Table 11 in the Appendix.

## 5 EXPERIMENTAL RESULTS

We evaluate our HiP3 along with state-of-the-art O2O RL and online RL algorithms across two categories of online tasks: the same online tasks and novel online tasks, based on the D4RL benchmark (Fu et al., 2020) (See Appendix A.9 for a brief introduction.). Our experiments mainly focus on the novel online tasks because adapting to novel online tasks is the inherent challenge of the O2O RL methods. We seek to address the following three questions: (1) **Comparison with prior methods:** Can HiP3 outperform state-of-the-art O2O RL algorithms across various tasks, particularly in terms of rapid adaptation to novel online tasks? (2) **Ablation study:** What is the optimal policy selection algorithm for HiP3? Moreover, how do the number of policies $K$, the corresponding selection method for Top-K policies, the number of reference vectors, and the presence of a policy pool affect HiP3's performance? (3) **Effectiveness of HiP3:** Is HiP3 capable of selecting the suitable policies at different states for diverse online tasks?

**To address question (1)**, we evaluate the performance of HiP3 in comparison with two distinct categories of algorithms: (i) O2O RL algorithms, including CAL-QL (Nakamoto et al., 2024), PEX (Zhang et al., 2023), AWAC (Nair et al., 2020), IQL (Kostrikov et al., 2021), CQL (Kumar et al., 2020), and FamO2O (Wang et al., 2024), and (ii) online RL algorithm: MBPO (Janner et al., 2019). While the implementations of CAL-QL, AWAC, IQL, and CQL are obtained from CORL (Tarasov et al., 2024), we utilize the official implementations of PEX and MBPO for analysis. To ensure a fair comparison, all the O2O RL algorithms (except FamO2O) are trained for 1M steps in the offline stage and 100K steps in the online stage, and the online RL algorithm MBPO is trained for 100K steps. Additional specifications of these baseline methods are provided in Appendix A.10.

As shown in Table 1, HiP3 consistently demonstrates state-of-the-art (SOTA) performance on the same online tasks, outperforming baselines in 7 out of 12 datasets. On the novel online tasks, which constitute our primary focus, HiP3 also attains SOTA performance, excelling in 8 out of 12 datasets. As evident in Figure 3a, existing representative O2O RL methods exhibit difficulties in consistently yielding satisfactory results across a majority of datasets for novel online tasks. This is attributed to the inherent fixed optimism-pessimism trade-off within these methods, which hinders their ability to effectively adapt to novel online tasks. Furthermore, as indicated in Figure 3b, even for certain identical online tasks, some prior O2O-RL methods fail to achieve satisfactory outcomes. This inadequacy may be attributed to their dependency on offline-pre-trained policies, which might not have been adequately learned on low-quality datasets such as random and medium-replay, thus resulting in sluggish performance improvements for these tasks. However, HiP3 can effectively addresses these limitation by offering a policy pool that includess policies with varied optimism-pessimism trade-offs. Furthermore, our hierarchical RL framework, based on contextual bandit algorithm, facilitates the efficient selection of suitable policies from the pool at different states for

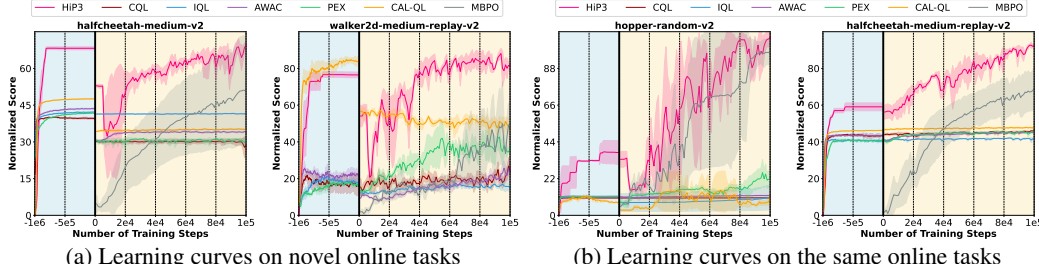

(a) Learning curves on novel online tasks          (b) Learning curves on the same online tasks

Figure 3: Offline-to-online learning curves on the same and novel online tasks. These curves are averaged over 3 random seeds (each with 10 evaluations), and the shaded area depicts the standard deviation of these runs. The results show that HiP3 outperforms the SoTA O2O RL methods (i.e., AWAC, CAL-QL, CQL, IQL, PEX) and online RL method (MBPO). Full result on all datasets can be found in Figure 7 and Figure 8 in the Appendix.

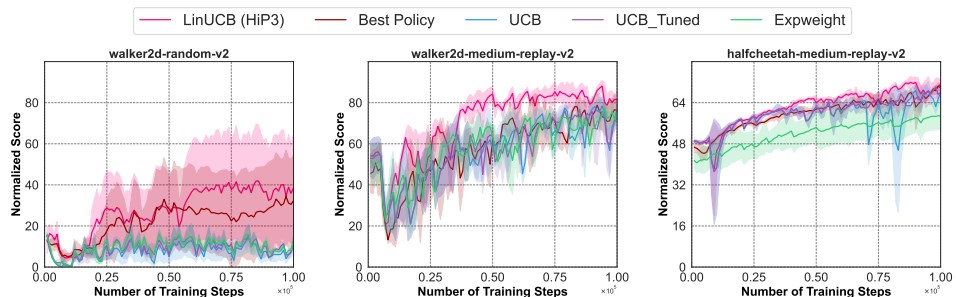

Figure 4: Ablation study on policy selection algorithms for novel online tasks. These curves indicate the performance during the online learning stage. LinUCB Chu et al. (2011) achieves the best performance. More results can be found in Figure 9 in the Appendix.

online fine-tuning. These findings underscore HiP3's efficacy on both the same and novel online tasks, demonstrating its capability to adapt to novel online tasks with limited interactions.

**To address question (2),** we conduct a thorough ablation study on novel online tasks, focusing on five key designs of HiP3: context bandit algorithm for policy selection, number of policies, $K$, in the pool, existence of policy pool, amounts of reference vectors and method for selecting the Top-K policies. Figure 4 shows the performance of HiP3 with various policy selection algorithms, including Best Policy, UCB Auer et al. (2002), UCB-Tuned Garivier and Cappé (2011), Expweight Kessler et al. (2022), and LinUCB Chu et al. (2011), the latter of which is incorporated in our HiP3 framework. As we can see, the state-dependent policy selection algorithm, LinUCB, consistently outperforms alternative policy selection algorithms in majority of the tasks. These findings suggest that relying on a solitary policy for online deployment is inadequate, likely due to the varying degrees of state similarity. For instance, a pessimistic policy might be suitable for similar states, whereas an optimistic policy could be more effective for exploring new states. Consequently, our state-dependent hierarchical policy selection algorithm grounded in LinUCB is more apt for online fine-tuning.

Figure 5 demonstrates the performance of HiP3 with differing numbers of bandit arms/policies, $K$, in the pool. Our findings indicate that the selection of an excessive number of policies can result in suboptimal performance during online fine-tuning since these policies may not receive adequate training with limited online interaction steps. In addition, choosing solely the best-performing policy based on evaluation is insufficient as it might fail to address the optimism-pessimism trade-off essential for effective online learning. Therefore, we adopt a top-$K$ selection strategy to choose the most appropriate policies for various online tasks. According to our ablation study, configuring the system with $K = 5$ delivers the best performance across most of the tasks.

Additionally, we ablate whether generating a policy pool during the offline stage affects online fine-tuning performance, the significance of diverse reference amounts, and the effect of using different methods for choosing Top-K policies, we find positive evidences for all these experiments. Corresponding results are reported in Figure 11, Table 9 and Table 10 in the Appendix due to space constraints.

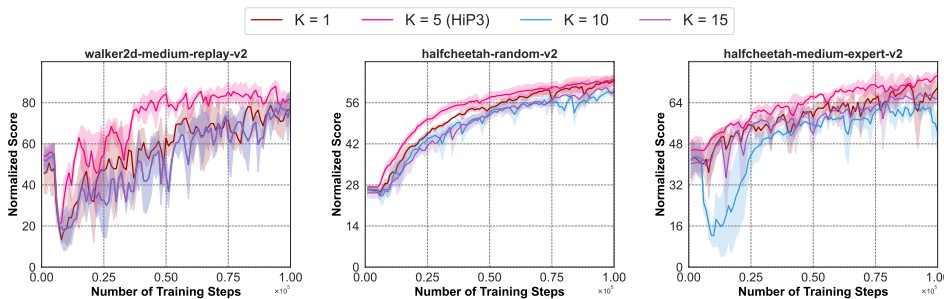

Figure 5: Ablation study on the number of policies $K$ when fine-tuned the policy pool on novel online tasks. These curves indicate the performance during the online learning stage. $K = 5$ achieves the best performance. More results can be found in Figure 10 in the Appendix.

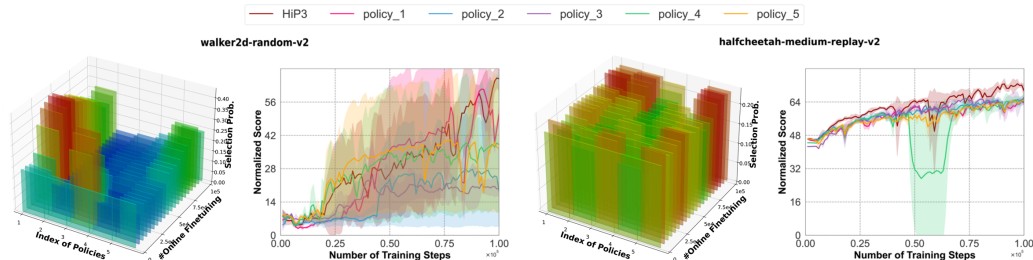

Figure 6: Effectiveness of HiP3. We evaluate the effectiveness of HiP3 by visualizing the probability distribution generated by the contextual bandit algorithm at each online interaction step, and we also provide the learning curve of HiP3 and the five policies in the pool. It is observed that our method converges to the policy that achieves the best fine-tuning performance on both novel online tasks. More results can be found in Figure 12 in the Appendix.

**To address question (3),** we conduct a comprehensive analysis of HiP3's policy selection process across varying states during the online fine-tuning on novel online tasks. As depicted in Figure 6, the heat maps illustrate the selection probabilities for each policy throughout the online RL stage, emphasizing the dynamic adaptation of these probabilities over time. This adaptability signifies HiP3's capability to optimize policies within the domain of online RL. Moreover, the learning curve associated with walker2d-random-v2 (where a single policy outperforms others) indicates HiP3's efficacy in identifying the most suitable policy for online adaptation. Notably, HiP3 occasionally exceeds the performance of the optimal policy, thereby demonstrating its proficiency in state-dependent policy selection. Regarding the learning curve of halfcheetah-medium-replay-v2 (where policies exhibit similar performance), it is observed that HiP3 can distribute policy selections equitably among similar performing policies, while still surpassing the performance of any single policy during online fine-tuning. Hence, it is evident that HiP3 tends to progressively concentrate selection probabilities towards specific policies as training proceeds. This convergence is crucial since it enables the algorithm to autonomously identify and stabilize upon the most appropriate policy for various states, thereby ensuring robust performance across a diverse array of tasks. Furthermore, the hierarchical RL framework of HiP3, which integrates policies with varying optimism-pessimism trade-offs, facilitates the efficient selection of suitable policies across states, enhancing its adaptability and efficacy in online fine-tuning.

## 6 CONCLUSION

This paper identifies the optimism-pessimism trade-off significantly affects the performance of model-based O2O RL. However, it is challenging for existing methods to balance this trade-off in a single objective. Instead, we develop a bi-objective formulation for the optimism-pessimism trade-off. The resulting algorithm, HiP3, produces a diverse pool of Pareto policies performing different levels of optimistic-pessimistic trade-offs. When fine-tuned on online tasks, the most suitable policy can be selected from the pool to fine-tune at each online interaction step. Empirically, our method outperforms prior online RL and O2O RL algorithms and sets a new state-of-the-art on the O2O RL benchmarks with diverse online tasks.

## 7 ETHICS STATEMENT AND REPRODUCIBILITY STATEMENT

This work focuses on methodological development and empirical evaluation of model-based offline-to-online reinforcement learning in standard benchmark environments (e.g., MuJoCo (Todorov et al., 2012) and D4RL (Fu et al., 2020)). All experiments are conducted on publicly available simulated datasets, and the procedures for creating novel online environments are provided with code. No human subjects or private data are involved. We do not foresee direct ethical concerns arising from our study. While reinforcement learning methods may influence future applications in robotics or other real-world systems, such broader impacts are beyond the scope of this work.

We have taken multiple steps to ensure the reproducibility of our work. The implementation details of HiP3, including model architectures, environments, hyper-parameters, are described in detail in Section 4, Appendix A.5, Appendix A.9 and Appendix A.13. In addition, we will release the source code upon publication to facilitate verification and further research. The datasets used in our experiments are publicly available through the D4RL benchmark.

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

# A APPENDIX

## A.1 USAGE OF LARGE LANGUAGE MODELS

Regarding the usage of Large Language Models (LLMs), they were employed only for minor language polishing of the manuscript and for generating auxiliary code snippets used in preliminary testing. All core ideas, algorithmic designs, and experimental results were solely developed and implemented by the authors.

## A.2 LIMITATIONS AND BROADER IMPACTS

While our HiP3 achieves the state-of-the-art performance on many O2O RL benchmarks, it still experiences initial performance degradation at the beginning of online fine-tuning (similar to other baseline methods). We have investigate it in Appendix A.14 and prepare to solve this problem in future work. Regarding its broader impacts, HiP3 is a fundamental research in RL and we anticipate no adverse societal consequences beyond those generally associated with RL.

## A.3 CONVERGENCE ANALYSIS

**Assumption 1.** Suppose the $m$ objectives $f_1, f_2, \ldots, f_m$ are differentiable and their gradients are $L_i$-Lipschitz continuous with constant $L_i > 0$.

**Assumption 2.** For any $i \in \{1, 2, \cdots, m\}$, $\{g_i(\theta_t)\}_{t=1}^{\infty}$ is an independent stochastic gradient sequence satisfying:

$$\mathbb{E}[g_i(\theta_t)] = \nabla f_i(\theta_t), \ Var(g_i(\theta_t)) = \mathbb{E}[||g_i(\theta_t) - \nabla f_i(\theta_t)||^2] \leq \sigma^2.$$

**MGDA Update:** In the ascent stage of MOSAIC (see Alg. 2), the policy is updated via a weighted combination of the three objective's gradients. Define the combined update direction

$$d_t = \sum_{i=1}^{m} \beta_{i,t} \ g_i(\theta_t),$$

where the weights $\beta_{i,t} \geq 0$ are chosen by solving the MGDA subproblem (Eq. 5), ensuring that no individual objective is decreased at that step (i.e., $\nabla f_i^\top d_t \geq 0, \forall i$). The stochastic policy update is then:

$$\theta_{t+1} = \theta_t - \eta d_t.$$

Let $\bar{d}_t = \mathbb{E}[d_t]$ denote the expected update direction.

**Theorem 1** (Convergence of MOSAIC). *Suppose that Assumption 1 and 2 hold. Let $0 < c < \frac{2}{L_i}$ and $\Delta_i = f_i(\theta_i^*) - f_i(\theta_0)$, where $\theta_i^*$ is the minimum of objective $f_i$. For any fixed $T$, define the constant stepsize $\eta$ as*

$$\eta = \frac{c}{\sqrt{T}}.$$

*Denote $\Delta = \max_i\{\Delta_i\}$. Then for any $\epsilon > 0$, after*

$$T = O\left(\frac{1}{\epsilon^2}\left(\frac{\Delta}{c} + c \, L_i \sigma^2\right)^2\right)$$

*iterations, we have:*

$$\frac{1}{T}\sum_{t=0}^{T-1} \mathbb{E}[||\bar{d}_t||^2] \leq \epsilon.$$

*This implies $||\bar{d}_t|| \to 0$ as $T \to \infty$, which is the necessary condition for convergence to a Pareto stationary solution.*

*Proof.* By $L_i$-smoothness, we have the following inequality for each $f_i$:

$$f_i(\theta_{t+1}) \le f_i(\theta_t) + \nabla f_i(\theta_t)^\top (\theta_{t+1} - \theta_t) + \frac{L_i}{2}\|\theta_{t+1} - \theta_t\|^2$$

$$= f_i(\theta_t) - \eta \, \nabla f_i(\theta_t)^\top \, d_t + \frac{L_i}{2}\eta^2\|d_t\|^2$$

Taking the conditional expectation $\mathbb{E}_t[\cdot]$ in both sides, we have

$$\mathbb{E}_t[f_i(\theta_{t+1})] \le f_i(\theta_t) - \eta \, \nabla f_i(\theta_t)^\top \, \bar{d}_t + \frac{L_i}{2}\eta^2 \mathbb{E}_t[\|d_t\|^2], \tag{6}$$

where we have used the fact $\bar{d}_t = \mathbb{E}[d_t]$.

Also note that

$$\mathbb{E}_t\left[\|d_t\|^2\right] = \|\mathbb{E}_t[d_t]\|^2 + \mathbb{E}_t\left[\|d_t - \mathbb{E}_t[d_t]\|^2\right]$$

$$= \|\bar{d}_t\|^2 + \mathbb{E}_t\left[\left\|\sum_{i=1}^m \beta_{i,t}\, g_i(\theta_t) - \mathbb{E}_t\left[\sum_{i=1}^m \beta_{i,t}\, g_i(\theta_t)\right]\right\|^2\right]$$

$$= \|\bar{d}_t\|^2 + \sum_{i=1}^m \mathbb{E}_t\left[\|\beta_{i,t}\, g_i(\theta_t) - \mathbb{E}_t[\beta_{i,t}\, g_i(\theta_t)]\|^2\right]$$

$$= \|\bar{d}_t\|^2 + \sum_{i=1}^m \beta_{i,t}^2\, \mathbb{E}_t\left[\|\, g_i(\theta_t) - \mathbb{E}_t[\, g_i(\theta_t)]\|^2\right] \tag{7}$$

$$\le \|\bar{d}_t\|^2 + \sigma^2 \sum_{i=1}^m \beta_{i,t}^2$$

$$\le \|\bar{d}_t\|^2 + \sigma^2 \left(\sum_{i=1}^m \beta_{i,t}\right)^2$$

$$= \|\bar{d}_t\|^2 + \sigma^2,$$

where we use the basic properties of variance of multiple independent random variables in the first 4 equalities; we use Assumption 2 in the first inequality; we use the fact

$$\sum_{i=1}^m \beta_{i,t}^2 \le \left(\sum_{i=1}^m \beta_{i,t}\right)^2$$

in the second inequality.

Substituting Eq. (7) into Eq. (6), we have

$$\mathbb{E}_t[f_i(\theta_{t+1})] \le f_i(\theta_t) - \eta \, \nabla f_i(\theta_t)^\top \, \bar{d}_t + \frac{L_i}{2}\eta^2 \left(\|\bar{d}_t\|^2 + \sigma^2\right)$$

$$\le f_i(\theta_t) - \eta \, \|\bar{d}_t\|^2 + \frac{L_i}{2}\eta^2 \left(\|\bar{d}_t\|^2 + \sigma^2\right),$$

where we use the fact $\nabla f_i^\top \bar{d}_t \ge \|\bar{d}_t\|^2$ (Lemma 2.1 in (Désidéri, 2012)) in the second inequality.

Taking the total expectation $\mathbb{E}[\cdot]$ in both sides and rearranging it, we have

$$\left(\eta - \frac{L_i}{2}\eta^2\right) \mathbb{E}[\|\bar{d}_t\|^2] \le \mathbb{E}\left[f_i(\theta_t) - f_i(\theta_{t+1})\right] + \frac{L}{2}\eta^2\sigma^2$$

Summing over $t = 0$ to $T - 1$ and dividing into $T$ in both sides, we have

$$\frac{1}{T}\sum_{i=0}^{T-1} \mathbb{E}\left[\|\bar{d}_t\|^2\right] \le \frac{f_i(\theta_0) - f_i(\theta^*)}{T\left(\eta - \frac{L_i}{2}\eta^2\right)} + \frac{L_i\,\eta^2\,\sigma^2}{2\left(\eta - \frac{L_i}{2}\eta^2\right)}.$$

For any fixed $T$, we can set $\eta = \frac{c}{\sqrt{T}}$, where $0 < c < \frac{2}{L}$, then we have

$$\frac{1}{T}\sum_{i=0}^{T-1}\mathbb{E}\left[\|\bar{d}_t\|^2\right] = O\left(\frac{\Delta_i}{c\sqrt{T}} + \frac{cL_i\sigma^2}{\sqrt{T}}\right).$$

Denote $\Delta = \max_i\{\Delta_i\}$. For any $\epsilon > 0$, after

$$T = O\left(\frac{1}{\epsilon^2}\left(\frac{\Delta}{c} + cL_i\sigma^2\right)^2\right)$$

iterations, we have

$$\frac{1}{T}\sum_{i=0}^{T-1}\mathbb{E}\left[\|\bar{d}_t\|^2\right] \le \epsilon.$$

$\square$

### A.4 UNCERTAINTY ESTIMATION $u$ IN HiP3

The predicted uncertainty $u$ of the environment model $M$ is estimated as the maximum standard deviation among the ensemble, i.e., $u(s_t, a_t) = \max_{i=1}^N \|\sigma_i(s_t, a_t)\|$. This ensemble-based uncertainty estimation strategy follows established practices in model-based offline RL such as MOPO (Yu et al., 2020) and MOReL (Kidambi et al., 2020). This method is favored for its scalability, ease of implementation, and effectiveness in representing epistemic uncertainty in deep neural networks. Although alternative techniques such as Bayesian methods(Lin et al., 2023) and unsupervised representation-based approaches(Pérez et al., 2023) are available, recent studies (Rahaman et al., 2021; Ghasemipour et al., 2022) have demonstrated that deep ensembles tend to provide more accurate and better-calibrated uncertainty estimates. These results indicate that ensemble methods strike an effective balance between predictive performance and computational cost, making them a reliable and practical choice for uncertainty estimation in HiP3.

A limitation of using predicted return and uncertainty directly is that they may not match the true dynamics in out-of-distribution regions, especially when the offline dataset has limited coverage. Conservative approaches (Yu et al., 2020; Rafailov et al., 2023) address this by penalizing high-uncertainty predictions. While effective in reducing overestimation, this requires careful tuning of penalty coefficients and may restrict exploration during online fine-tuning.

HiP3 avoids the use of fixed penalties by casting return and uncertainty as two separate objectives in a bi-objective optimization problem. This formulation produces a Pareto set of offline policies with different optimism–pessimism levels. During online fine-tuning, a hierarchical selection mechanism adaptively chooses among these policies according to the current state, which improves both safety and efficiency of adaptation.

### A.5 ENVIRONMENT MODEL $M$

The environment model used by HiP3 is developed by a bootstrap ensemble of $N$ environment models: $M = \{M_i\}_{i=1}^N$. Each member $i$ of the ensemble is a probabilistic neural network, taking state-action pairs $(s_t, a_t)$ as input and outputting a Gaussian distribution over the next state and reward: $\hat{P}_i(s_{t+1}, r|s_t, a_t) = \mathcal{N}(\mu_i(s_t, a_t), \sigma_i(s_t, a_t))$. The predicted reward $\hat{r}$ of the environment model $M$ is the mean of the rewards among the ensemble, i.e., $\hat{r}(s_t, a_t) = \text{mean}_{i=1}^N \mu_i(s_t, a_t)$, and the predicted uncertainty $u$ of the environment model $M$ is estimated as the maximum standard deviation among the ensemble, i.e., $u(s_t, a_t) = \max_{i=1}^N \|\sigma_i(s_t, a_t)\|$.

This environment model is trained according to the design of MOPO (Yu et al., 2020). Specifically, we trained an ensemble of 7 models, from which the 5 best models ($N = 5$) in terms of the prediction errors are selected. Each model of the ensemble is independently trained using the maximum likelihood estimation with mini-batch stochastic gradient descent (Jain et al., 2018). During the inference stage, the policy interacts with a randomly selected model from the $N$ ensemble models, generating state-reward pairs from the corresponding distributions. This approach introduces variability by sampling different transitions from different models within a single episode, leading to a highly stochastic environment model. Previous works (Janner et al., 2019; Yu et al., 2020) have

shown that this environment model effectively mitigates the model exploitation problem in offline RL scenarios. We maintain the same hyper-parameters for our environment model as those used in MOPO for the MuJoCo tasks.

The hyper-parameters of the environment model $M$ are provided in Table 2. For the MuJoCo tasks, we adopt the same hyper-parameter settings as those used in MBPO (Janner et al., 2019) given their proven effectiveness in similar continuous control environments.

| Hyper-parameter | Value |
|---|---|
| Number of models $N$ | 5 |
| Model architecture | FC(200, 200, 200, 200) |
| Batch size | 256 |
| Optimizer | Adam |
| Learning rate | $1 \times 10^{-3}$ |
| Holdout ratio | 0.2 |

Table 2: The hyper-parameters used for environment model $M$

### A.6 MULTI-OBJECTIVE OPTIMIZATION

Multi-objective optimization (MOO) refers to the process of optimizing two or more (potentially conflicting) objective functions simultaneously (Deb et al., 2002). Formally, given a decision space $\mathcal{X}$ and a set of $d$ objective functions $f_i : \mathcal{X} \to \mathbb{R}$ for $i \in \{1, \cdots, d\}$, the goal is to find a solution $\boldsymbol{x} \in \mathcal{X}$ that balances the trade-offs among objectives:

$$\max_{\boldsymbol{x} \in \mathcal{X}} \mathbf{f}(\boldsymbol{x}) \triangleq \Big( f_1(\boldsymbol{x}), f_2(\boldsymbol{x}), \cdots, f_d(\boldsymbol{x}) \Big).$$

The trade-offs among objectives arise because improving one objective may result in degradation in another as there may not exist a single solution that attains the maximums across all objectives.

Different from single-objective optimization, where a unique optimal solution often exists, in MOO there always exists a set of solutions with diverse trade-offs. These solutions are called **Pareto optimal**, and the corresponding **Pareto front** is formed by the set of these solutions, whose definitions are provided below.

**Dominance**    Given two solutions $\boldsymbol{x}_1 \in \mathcal{X}$, $\boldsymbol{x}_2 \in \mathcal{X}$, and corresponding objectives $f_i, i \in \{1, \ldots, d\}$, we say that $\boldsymbol{x}_1$ *dominates* $\boldsymbol{x}_2$ (denoted $\boldsymbol{x}_1 \succ \boldsymbol{x}_2$) if and only if:

$$\forall i \in \{1, \cdots, d\}, \ f_i(\boldsymbol{x}_1) \geq f_i(\boldsymbol{x}_2) \quad \text{and} \quad \exists j \in \{1, \cdots, d\}, \ f_j(\boldsymbol{x}_1) > f_j(\boldsymbol{x}_2).$$

**Pareto Optimal**    A solution $\boldsymbol{x}^*$ is said to be *Pareto optimal* if there does not exist another solution $\boldsymbol{x}'$ such that the $\boldsymbol{x}' \succ x^*$.

**Pareto Front**    The *Pareto front* is the set of all Pareto optimal solutions:

$$\mathcal{P} = \{\boldsymbol{x} \mid \nexists \ \boldsymbol{x}' \text{ such that } \boldsymbol{x}' \succ \boldsymbol{x}\} .$$

In real-world applications, there always exist some scenarios that involve conflicting objectives. By providing a diverse set of solutions along the Pareto front, MOO enables a greater flexibility in selecting a proper solution for specific task requirements.

### A.7 COMPARISON WITH LINEAR SCALARIZATION METHOD

There also exist some multi-objective RL papers, particularly DIME (Abdolmaleki et al., 2021). These methods tend to solve the multi-objective problem with a Linear Scalarization (LS) $r - \lambda u$ method. LS methods are indeed simpler than the HiP3 framework. However, we do not choose the LS method for the offline phase because, in the multi-objective community, it is well-known that

LS cannot guarantee to find solutions on the non-concave parts of the Pareto front for maximization problems (Boyd and Vandenberghe, 2004).

Moreover, to further demonstrate that HiP3 outperforms LS methods, we also combined our bandit method with DiME (Abdolmaleki et al., 2021). Due to the lack of open-source implementation, we built the baseline on top of AWAC based on the details provided in the original paper. As shown in Table 3, DiME+AWAC outperforms the vanilla AWAC on several novel online tasks, indicating the efficacy of DiME and our bandit method, but still underperforms HiP3 under the same computation budget constraint (1M training steps for the offline phase and 100K steps for the online phase). The result supports the motivation of Eq. 2: applying MGDA-style optimization to the proposed bi-objective formulation enables more efficient coverage of the Pareto front at the offline stage and better online adaptation to novel tasks.

| Method | HiP3 | DiME+AWAC | AWAC |
|---|---|---|---|
| halfcheetah-random-v2 | 63.8 | 28.4 | 33.2 |
| halfCheetah-medium-replay-v2 | 72.0 | 36.0 | 29.9 |
| hopper-random-v2 | 89.7 | 7.0 | 9.2 |
| hopper-medium-replay-v2 | 92.3 | 51.2 | 29.6 |
| mean | 79.5 | 30.7 | 25.5 |

Table 3: Comparison with Linear Scalarization Method

## A.8 BANDIT ALGORITHM

A bandit problem (Lattimore and Szepesvári, 2020) is a sequential game between a learner and an environment. In each round $t$ of the game, the learner chooses an arm $A_t$ from a given set of arms $\mathcal{A}$, and the environment returns a corresponding reward $r_t$. The key challenge of bandit problem is that the environment is unknown to the learner, so the learner can only choose the arm based on the history $H_{t-1} = (A_1, r_1, \cdots, A_{t-1}, r_{t-1})$.

In the standard formulation of multi-armed bandit, each arm remains consistent throughout training. However, this static approach does not suit HiP3, where policy optimization is essential for improved fine-tuning performance. Instead, we adopt a formulation where each arm is modeled as an independent Markov machine (Whittle, 1988). When a particular arm is played, the state of that machine advances to a new one, and the reward also depends on the current state of that machine.

A variation of this problem, the contextual bandit (Chu et al., 2011), considers both the selected arm and the context in which it is chosen. By modeling our hierarchical RL framework based on the contextual bandit, we can efficiently utilize the context (state) information to select the appropriate arm (policy) in a state-dependent way.

In our hierarchical RL framework, each policy in the Pareto policy pool is treated as an arm of the bandit, and we adopt LinUCB (Chu et al., 2011) as the policy selection algorithm for the bandit $B(A, b)$ with $K$ arms, where $A$ is initialized as a collection of $K$ $d$-dimensional identity matrices, and $b$ is initialized as a collection of $K$ $d$-dimensional zero vectors. At each online interaction step $t$, LinUCB calculates the probability distribution over the arms/policies according to the current state $s_t$ with the highest likelihood one is chosen:

$$P \leftarrow Softmax((A^{-1}b)^T s_t + \omega\sqrt{s_t^T A^{-1} s_t}),$$
$$\pi'_\theta \leftarrow \arg\max(P), \tag{8}$$

where $\omega$ is a hyper-parameter for confidence bound. The selected policy interacts with the online environment to receive the corresponding reward $r_t$, which is subsequently used to update the LinUCB's parameters $(A, b)$:

$$A \leftarrow A + \gamma s_t s_t^T,$$
$$b \leftarrow b + \gamma s_t r_t, \tag{9}$$

where $\gamma$ represents the learning rate of the LinUCB algorithm, which is an adjustment introduced in comparison to prior implementations of LinUCB, and is designed to mitigate the risk of overfitting. The iterative learning procedure continues until certain convergence criteria are satisfied.

## A.9 MuJoCo Simulation Environment

We conduct the experiments on continuous control tasks using the MuJoCo (Todorov et al., 2012) simulation environment. For offline pre-training, we use the standard D4RL Gym benchmark (Fu et al., 2020), which provides datasets with diverse qualities. Our HiP3 and all the baseline methods are tested on three tasks: Walker2d, Hopper, and HalfCheetah. For each task, these algorithms are evaluated on datasets with four different quality levels: random-v2, medium-v2, medium-replay-v2, and medium-expert-v2. These datasets are generated as follows. "Random" is generated by collecting 1 million steps of rollouts from a randomly initialized policy. "Medium" is generated by collecting 1 million steps of rollouts from a partially trained SAC. "Medium-replay" is generated by recording all samples in the replay buffer observed during training until the policy reaches a medium level of performance. "Medium-expert" is generated by mixing 1 million steps of expert-level data with 1 million steps of sub-optimal data.

In the online stage, we categorize the experiments into two types of online tasks: the same online tasks and novel online tasks. For the former, the same offline task environments are used for the online tasks, while for the latter we introduce changes in both dynamics and reward structure, following standard practice in O2O-RL robustness evaluation. Concretely:

- Gravity increased from 9.8 -> 11.5

- Friction coefficients scaled by × 1.2

- Forward-reward weight decreased (0.85 of original)

- Control-cost weight increased to 0.2

- Noise scale is set (0.2 / 0.1 / 0.05 respectively for HalfCheetah, Hopper, Walker2d)

Together, these shifts produce a meaningful and measurable change in task difficulty, which is hard for current offline-to-online RL methods to maintain their performance.

## A.10 Implementation Details

We start by detailing our baseline implementations. For CAL-QL (Nakamoto et al., 2024), AWAC (Nair et al., 2020), IQL (Kostrikov et al., 2021), and CQL (Kumar et al., 2020), these algorithms were tested in an offline-to-online (O2O) settings provided by CORL (Tarasov et al., 2024), utilizing CORL's official implementation for baseline comparison. For MBPO (Janner et al., 2019) and PEX (Zhang et al., 2023), we utilized the official implementations provided by their authors. When conducting the comparison for PEX, we employed the same O2O setting provided by the official code. For MBPO, we learned the policies from scratch with the hyper-parameters provided by the official implementation. For FamO2O (Wang et al., 2024), we have referenced their published results, which involved running 1M steps for online fine-tuning.

To ensure a fair comparison, all the O2O RL algorithms (except FamO2O) were re-run from offline stage to online stage, i.e., 1M steps in the offline stage and 100K steps in the online stage to evaluate their performance with few-shot interactions. For MBPO, an online RL algorithm, we ran it for 100K steps in the online stage from scratch.

## A.11 Computing Infrastructure

Given the large number of experiments, our experiments were conducted using a mix of GPUs, including NVIDIA RTX-3090, NVIDIA V100, NVIDIA A100, and NVIDIA A6000. NVIDIA RTX-3090 was run on a Ubuntu operating system, while the other GPUs used CentOS. The software libraries and frameworks utilized in our experiments are detailed in the "readme.md" file provided in the source code zip file.

## A.12 COMPUTING COST

As shown in the Table 4 and Table 5 below, HiP3 takes comparable and in some cases even less training time than other offline-to-online RL methods at both offline and online stages (all experiments are conducted using a NVIDIA's A6000 GPU, and as for lack of time, we calculate average computing time with 3 seeds of random dataset and 3 seeds of medium-replay dataset).

While HiP3 needs to optimize a pool of Pareto policies in the offline stage, it doesn't take additional computation cost. For a fair comparison, we adopt the same number of training steps and samples as all baselines. In addition, to obtain more Pareto policies without linearly increasing the cost, we propose a warm-start strategy that leverages the achieved policies as initialization to find more Pareto optimal policies in their neighborhoods.

At the online stage, HiP3 only selects one single policy to optimize using the bandit algorithm at each RL step/phase, which means that HiP3 has the same and even less online training cost than other model-based RL baselines such as MBPO, given the negligible computation introduced by the bandit algorithm LinUCB. It only maintains a matrix $A \in \mathbb{R}^{n \times d \times d}$ and a vector $b \in \mathbb{R}^{n \times d}$, where $n < 10$ is the number of candidate policies and $d < 50$ is the dimension of the observation.

| Method | HiP3 (offline) | Cal-QL (offline) | PEX (offline) | MOPO (offline) |
|---|---|---|---|---|
| Hopper | 7.5h | 9.2h | 2.9h | 5.9h |
| HalfCheetah | 8.3h | 8.7h | 4.0h | 7.0h |
| Walker2d | 5.6h | 7.5h | 2.7h | 5.2h |

Table 4: Computation cost comparison for offline stage

| Method | HiP3 (online) | MBPO (online) |
|---|---|---|
| Hopper | 7.7h | 7.8h |
| HalfCheetah | 8.5h | 8.9h |
| Walker2d | 9.4h | 9.7h |

Table 5: Computation cost comparison for online stage

## A.13 THE HYPER-PARAMETERS OF HIP3

For the training of HiP3 on the MuJoCo tasks, the hyper-parameters for the offline and the online stages can be found in Table 6(a) and Table 6(b), respectively.

## A.14 PERFORMANCE DROP ANALYSIS

We believe the performance drop at the early phase of online fine-tuning is primarily caused by the gap between the offline-learned environment model and true environment dynamics. This challenge is shared across model-based offline-to-online RL methods. In our experiments, HiP3 demonstrates strong robustness on novel online tasks, showing better performance than MOPO+MBPO and Cal-ql (see Table 7). On the same online tasks (Table 8), Cal-ql exhibits the smallest average performance drop. However, HiP3 still maintains superior performance compared with MOPO+MBPO. The performance drop is notably mitigated in challenging datasets with low-quality offline data, such as random-v2 and medium-replay-v2. In future work, we plan to incorporate more advanced pretrained world models (e.g., NVIDIA's Cosmos world model (Agarwal et al., 2025)), which may further reduce this initial drop and improve adaptation efficiency.

## A.15 FULL EXPERIMENTAL RESULTS

The full experimental results are provided in this section. Figure 7 and Figure 8 illustrate the learning curves for baseline comparison when fine-tuning on the same and novel online tasks, respectively.

| Hyper-parameter | Value |
|---|---|
| Number of hidden units | 256 |
| Number of hidden layers | 2 |
| Optimizer | Adam |
| Rollout length | Walker2d: 1 |
| | HalfCheetah: 3 |
| | Hopper: 5 |
| Rollout batch size | 50 |
| Updates per step | 1 |
| Autotune SAC $\alpha$ | True |
| SAC $\alpha$ learning rate | $3 \times 10^{-4}$ |
| Max SAC $\alpha$ | 1.0 |
| Learning rate | $3 \times 10^{-4}$ |
| Discount factor | 0.99 |
| Constraint threshold $\psi$ | $-0.999$ |
| Target update rate | $5 \times 10^{-3}$ |
| Batch size | 256 |
| Number of reference vectors | 5 |
| Environment steps | $5 \times (180K + 2 \times 10K)$ |
| $(\tau_a, \tau_b)$ of reference vectors | $(0.1, 0.9)$ |
| Number of offline policies | 100 |

(a) Hyper-parameters for offline RL

| Hyper-parameter | Value |
|---|---|
| Number of hidden units | 256 |
| Number of hidden layers | 2 |
| Optimizer | Adam |
| Rollout length | 1 |
| Rollout batch size | $5 \times 10^4$ |
| Train model frequency | 500 |
| Model retain epoch | 5 |
| Updates per step | Walker2d: 20 |
| | HalfCheetah: 15 |
| | Hopper: 15 |
| Autotune SAC $\alpha$ | True |
| SAC $\alpha$ learning rate | $3 \times 10^{-4}$ |
| Max SAC $\alpha$ | 1.0 |
| Learning rate | $3 \times 10^{-4}$ |
| Discount factor | 0.99 |
| Target update rate | $5 \times 10^{-3}$ |
| Batch size | 256 |
| Number of epochs | 100 |
| Environment steps per epoch | 1000 |
| Number of online policies | 5 |

(b) Hyper-parameters for online RL

Table 6: The hyper-parameters of HiP3 used for offline and online learning on the MuJoCo tasks.

| Method | HiP3 | MOPO+MBPO | Cal-ql |
|---|---|---|---|
| halfcheetah-random-v2 | 8.3 | 18.7 | 17.2 |
| halfCheetah-medium-replay-v2 | 7.6 | 15.0 | 11.6 |
| hopper-random-v2 | 14.6 | 2.8 | 3.4 |
| hopper-medium-replay-v2 | 20.4 | 28.5 | 21.5 |
| mean | 12.7 | 16.3 | 13.4 |

Table 7: Performance drop comparison for novel online tasks

Figure 9 presents an ablation study on the policy selection algorithms used in the novel online tasks. Figure 10 explores the impact of the number of policies $K$ during fine-tuning on the novel online tasks. Figure 11 illustrates the importance of policy pool for online fine-tuning. Table 9 shows the significance of number of reference vectors. Table 10 illustrates the robustness of HiP3 algroithm with diverse methods of choosing Top-K policies. Figure 12 demonstrates the effectiveness of HiP3 by visualizing the probability distribution generated by the contextual bandit algorithm at each online interaction step. Finally, Table 11 presents comprehensive results of HiP3 as compared to seven baseline methods on 12 MuJoCo tasks, demonstrating the superior performance of HiP3 both on the same and novel online tasks.

| Method | HiP3 | MOPO+MBPO | Cal-ql |
|---|---|---|---|
| halfcheetah-random-v2 | 1.2 | 3.6 | 2.1 |
| halfCheetah-medium-replay-v2 | 1.4 | 9.8 | 4.8 |
| hopper-random-v2 | 12.1 | 1.5 | 4.2 |
| hopper-medium-replay-v2 | 16.7 | 23.1 | 2.6 |
| mean | 7.9 | 9.5 | 3.4 |

Table 8: Performance drop comparison for same online tasks

Figure 11 illustrates the importance of maintaining a policy pool. Without the policy pool, the offline phase essentially reduces to MOPO (Yu et al., 2020), which can be regarded as a variant of MBPO+MOPO. Our results suggest that training only a single policy in the offline phase limits the ability to adapt to diverse online environments and often yields suboptimal performance after pretraining. To address this limitation, HiP3 constructs a policy pool containing policies with varying levels of optimism–pessimism trade-offs during the offline stage. The ablation results confirm that this mechanism not only improves offline policy quality but also enhances adaptability to novel online tasks.

Table 9 reports the effect of varying the number of reference vectors. For fairness, we fix the total training budget to 1M steps and allocate it evenly across reference vectors. When the number of reference vectors is too large, each policy receives insufficient training, leading to unsatisfied performance. Conversely, too few reference vectors reduce the diversity of the policy pool. The ablation study shows that using five reference vectors achieves the best balance between policy diversity and overall performance.

Table 10 illustrates the robustness of HiP3 algorithm with diverse methods of choosing Top-K policies. HiP3 performs Top-K (K=5) policy selection by evaluating each policy in the Pareto pool on the online environment for a relatively small number n of rollouts with a truncated horizon h (We empirically found that setting and is sufficient to achieve the reported state-of-the-art performance on MuJoCo tasks. This incurs significantly lower computational cost compared to FQE, which requires training a separate critic network for each policy.), and then selecting the K policies having the highest cumulative rewards. More importantly, we acknowledge that this approach assumes access to online rollouts, which may be infeasible in safety-critical scenarios such as recommendation systems or autonomous driving. Hence, offline policy evaluation methods such as OPE (Le et al., 2019) or FQE (Jin et al., 2021) can be employed, and we found that "HiP3+FQE" only slightly degrades from "HiP3+online policy eval" but still achieves consistent improvements over the baselines on several novel online tasks

| Reference number | ref_vec=5 (HiP3) | ref_vec=4 | ref_vec=2 |
|---|---|---|---|
| halfcheetah-random-v2 | 63.8 | 63.3 | 63.5 |
| halfCheetah-medium-replay-v2 | 72.0 | 67.7 | 68.4 |
| hopper-random-v2 | 89.7 | 86.1 | 87.5 |
| hopper-medium-replay-v2 | 92.3 | 90.4 | 93.8 |
| mean | 79.5 | 76.8 | 78.3 |

Table 9: Ablation study for different number of reference vectors

| Method | Online Evaluation (HiP3) | FQE | PEX | Cal-ql |
|---|---|---|---|---|
| halfcheetah-random-v2 | 63.8 | 58.7 | 42.2 | 11.1 |
| halfCheetah-medium-replay-v2 | 72.0 | 67.8 | 30.0 | 32.6 |
| hopper-random-v2 | 89.7 | 87.1 | 23.5 | 17.1 |
| hopper-medium-replay-v2 | 92.3 | 100.1 | 39.4 | 65.9 |
| mean | 79.5 | 78.4 | 33.8 | 31.7 |

Table 10: Ablation study on method for choosing Top-K policies on novel online tasks

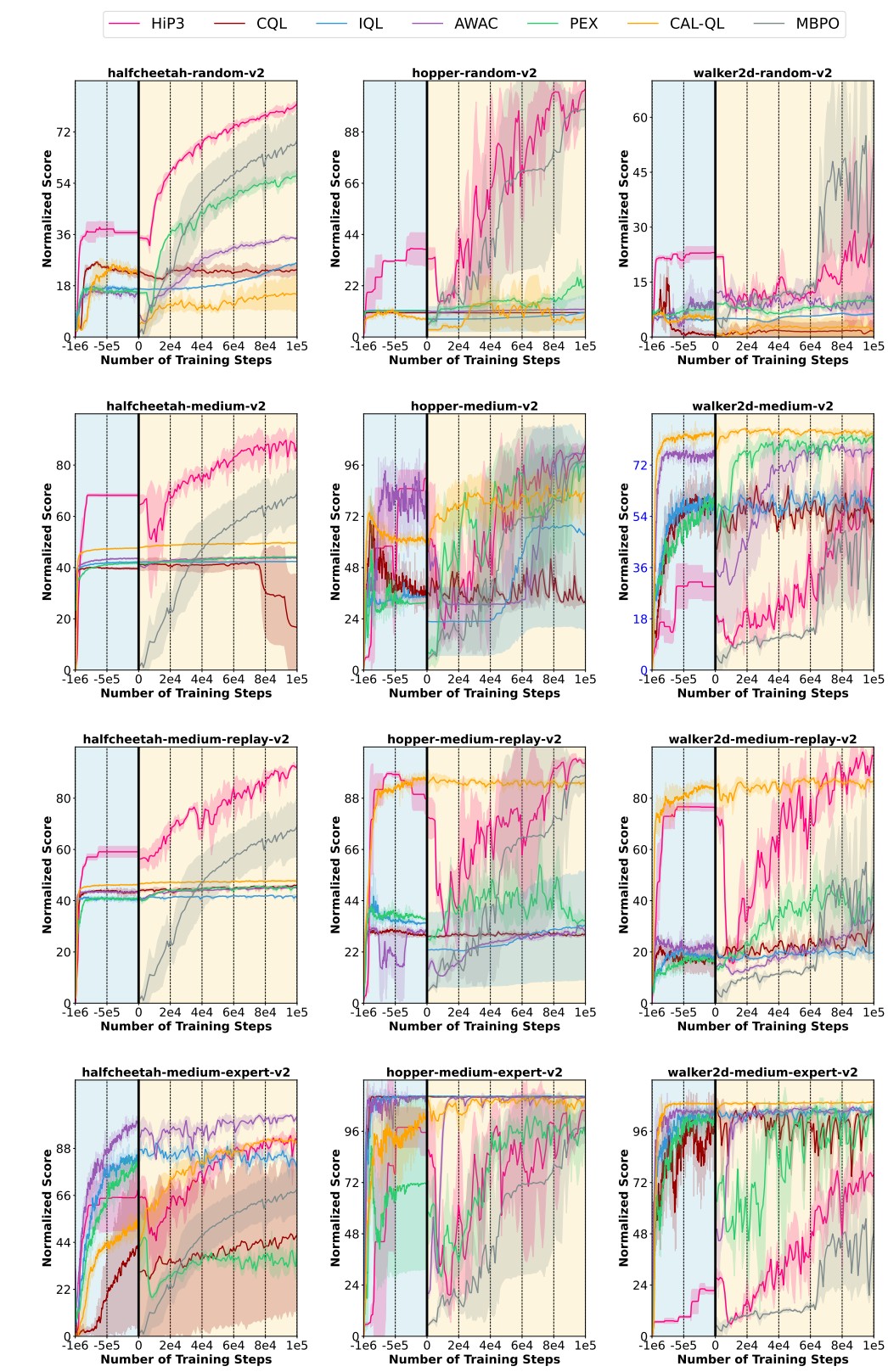

Figure 7: Offline-to-online learning curves on **the same online tasks**. These curves are averaged over 3 random seeds (each with 10 evaluations), and the shaded area depicts the standard deviation of these runs. The results show that HiP3 outperforms the SoTA O2O RL methods (i.e., AWAC, CQL, IQL, PEX) and the online RL method (MBPO) in majority of the same tasks.

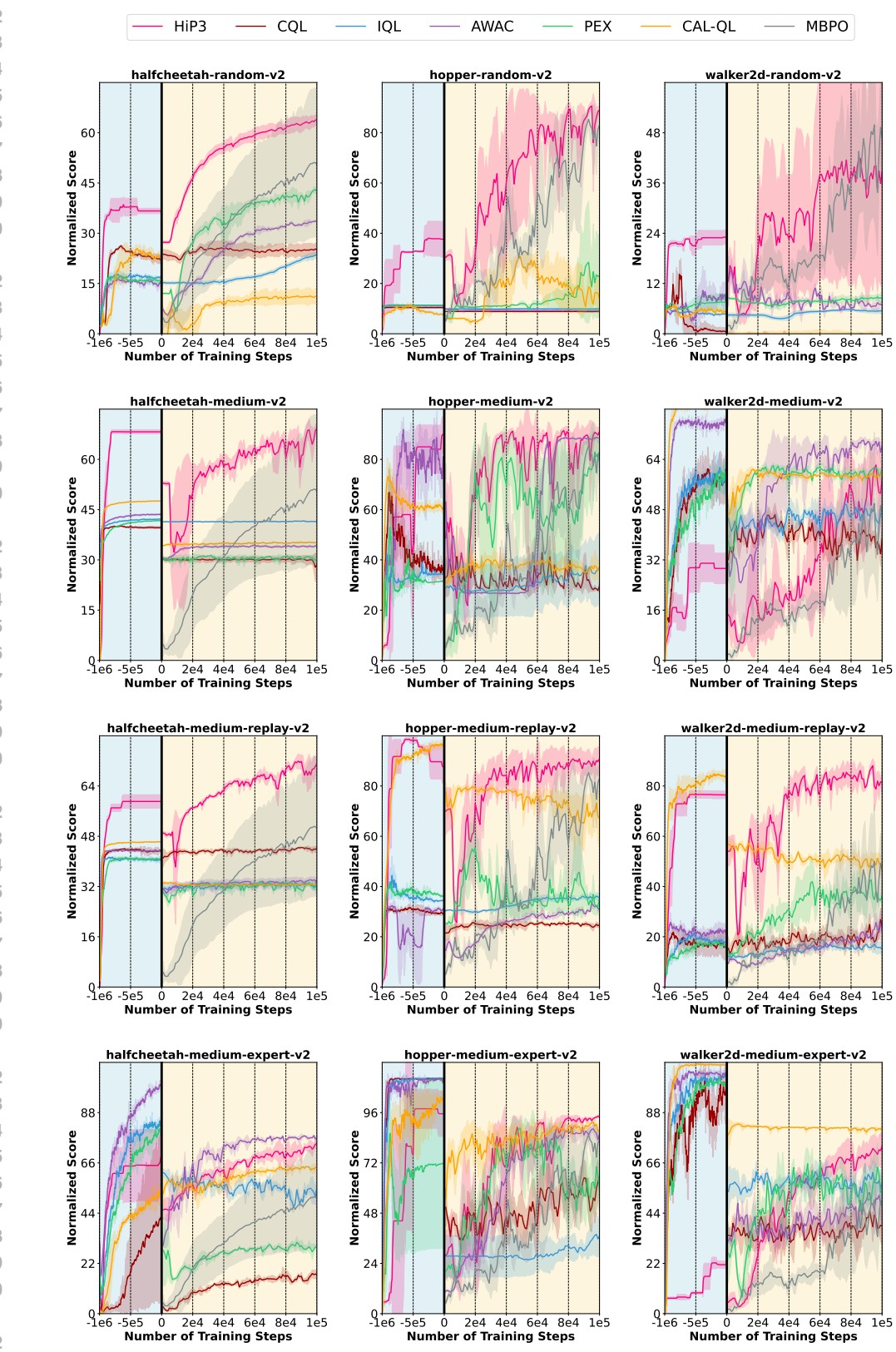

Figure 8: Offline-to-online learning curves on **the novel online tasks**. These tasks are generated via modifying the gravity, noise scales, and reward functions of the original MuJoCo tasks. These learning curves are averaged over 3 random seeds (each with 10 evaluations), and the shaded area depicts the standard deviation of these runs. The results show HiP3 outperforms the SoTA O2O RL methods (AWAC, CQL, IQL, PEX) and online RL method (MBPO) in majority of the novel tasks.

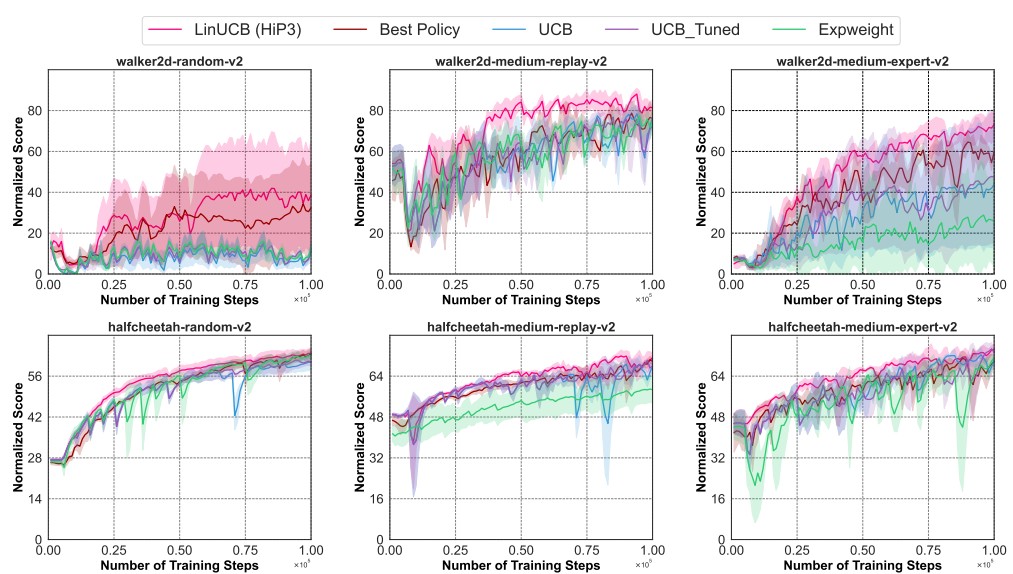

Figure 9: Ablation study on the policy selection algorithms for six novel online tasks. The normalized score is averaged over 3 random seeds (each with 10 evaluations). The curves exclusively illustrate the performance during the online stage, where various bandit algorithms are evaluated.

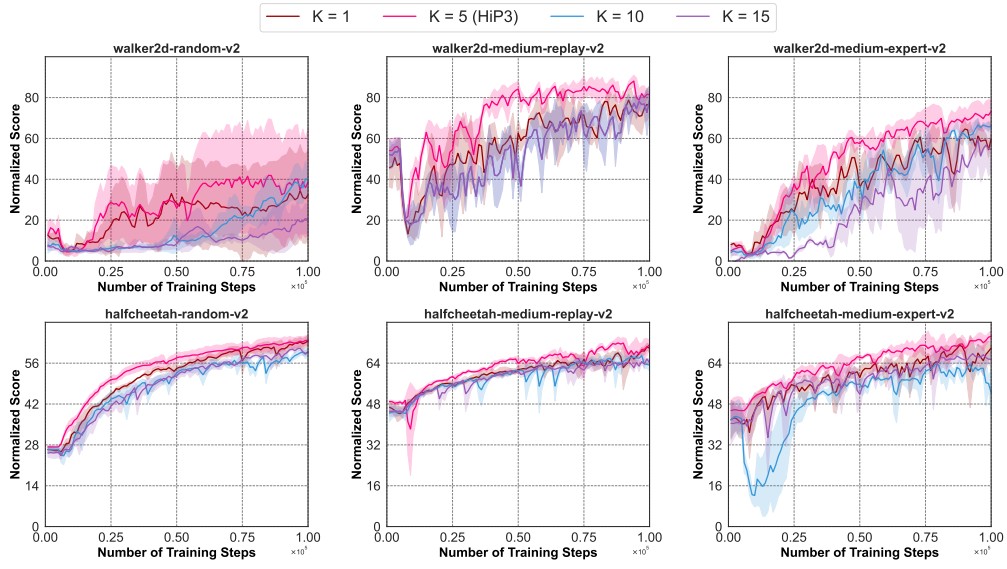

Figure 10: Ablation study on the number of policies $K$ when fine-tuning the policy pool on six novel online tasks. The normalized score is averaged over 3 random seeds (each with 10 evaluations). The curves exclusively illustrate the performance during the online stage, where LinUCB is used for policy selection from the pool.

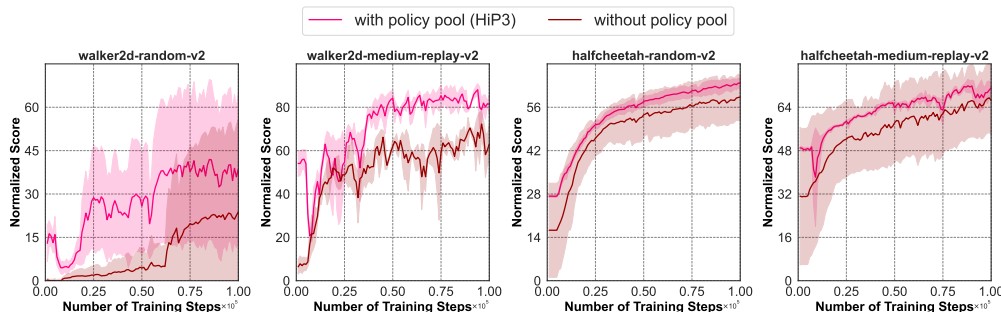

Figure 11: Ablation study on the impact of having a policy pool for online fine-tuning, in which a baseline method (w/o a policy pool) is compared with HiP3. This baseline method is close to MOPO (Yu et al., 2020), where a single policy is trained offline with SAC and fine-tuned online. The normalized score is averaged over 3 random seeds (each with 10 evaluations). The results show that constructing a policy pool during offline stage not only improves offline policy performance but also enhances adaptability to novel online tasks.

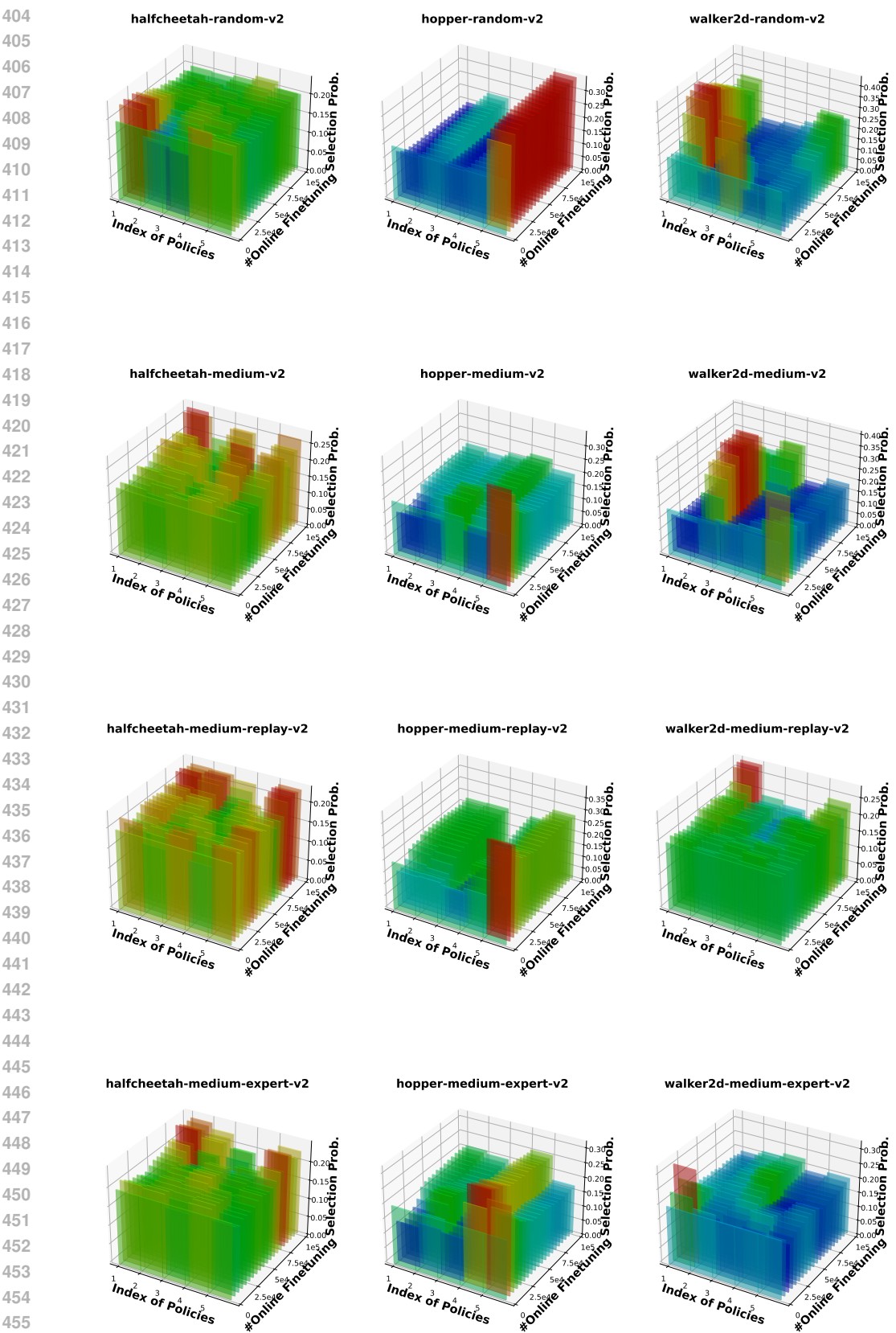

Figure 12: Effectiveness of HiP3. We evaluate the effectiveness of HiP3 by visualizing the probability distribution generated by the contextual bandit algorithm at each online interaction step.

|  |  | CQL | IQL | FamO2O* | AWAC | PEX | CAL-QL | MBPO | HiP3 |
|---|---|---|---|---|---|---|---|---|---|
| Random | HalfCheetah | 23.7 ±2.1 | 26.4 ±0.1 | – | 35.6 ±0.5 | 56.9 ±1.2 | 15.4 ±5.6 | 70.3 ±12.7 | 82.8 ±1.1 |
|  | Hopper | 10.5 ±0.1 | 10.4 ±7.3 | – | 11.7 ±0.2 | 22.2 ±4.1 | 12.7 ±8.5 | 98.9 ±6.0 | 107.5 ±2.8 |
|  | Walker2d | 1.7 ±2.7 | 6.3 ±0.2 | – | 12.2 ±6.7 | 9.8 ±2.2 | 1.6 ±1.3 | 52.4 ±26.9 | 30.2 ±25.6 |
| Medium | HalfCheetah | 16.7 ±20.7 | 42.4 ±0.4 | 53.4 | 43.6 ±0.8 | 44.1 ±0.3 | 49.7 ±0.3 | 70.3 ±12.7 | 89.1 ±5.1 |
|  | Hopper | 33.6 ±3.2 | 65.4 ±45.7 | 82.8 | 101.5 ±0.4 | 99.4 ±1.2 | 83.3 ±12.3 | 98.9 ±6.0 | 107.7 ±1.6 |
|  | Walker2d | 49.1 ±8.2 | 56.0 ±10.2 | 82.8 | 79 ±1.7 | 72 ±1.2 | 83.1 ±2.2 | 52.4 ±26.9 | 70.4 ±17.5 |
| Medium-replay | HalfCheetah | 46 ±0.1 | 41.5 ±0.5 | 51.0 | 45.3 ±0.1 | 45.9 ±0.1 | 47.8 ±0.1 | 70.3 ±12.7 | 92.5 ±0.9 |
|  | Hopper | 30.5 ±0.9 | 33.9 ±23.6 | 92.2 | 33.9 ±1.4 | 36 ±1.4 | 95.1 ±1.4 | 98.9 ±6.0 | 102.9 ±2.1 |
|  | Walker2d | 33.8 ±3.6 | 21.3 ±1.0 | 88.6 | 34.5 ±8.4 | 40.6 ±7.3 | 85.5 ±4.9 | 52.4 ±26.9 | 96.4 ±12.5 |
| Medium-expert | HalfCheetah | 48.4 ±31.8 | 76.7 ±6.8 | 91.8 | 103.9 ±1.5 | 48.3 ±5.6 | 92.5 ±1.6 | 70.3 ±12.7 | 91.1 ±3.4 |
|  | Hopper | 111.4 ±0.6 | 112.4 ±0.1 | 90.1 | 112.3 ±0.4 | 89.0 ±31.6 | 112.2 ±0.2 | 98.9 ±6.0 | 105.7 ±5.7 |
|  | Walker2d | 93.5 ±14.9 | 105.1 ±2.9 | 110.6 | 105.8 ±1.8 | 108.1 ±0.5 | 109.6 ±0.7 | 52.4 ±26.9 | 76.5 ±12.8 |
| Total Mean |  | 38.1 ±7.4 | 49.8 ±8.2 | 82.6 | 60.0 ±2.0 | 53.5 ±4.7 | 65.7 ±3.3 | 73.9 ±15.2 | 87.7 ±8.4 |

(a) Fine-tuning performance on the same online tasks

|  |  | CQL | IQL | AWAC | PEX | CAL-QL | MBPO | HiP3 |
|---|---|---|---|---|---|---|---|---|
| Random | HalfCheetah | 24.7 ±1.1 | 23.1 ±1.3 | 33.2 ±0.4 | 42.2 ±2.6 | 11.1 ±2.5 | 50.0 ±21.3 | 63.8 ±1.8 |
|  | Hopper | 8.9 ±0.1 | 9.9 ±0.0 | 9.2 ±0.1 | 23.5 ±17.2 | 11.7 ±4.7 | 85.1 ±3.8 | 89.7 ±2.7 |
|  | Walker2d | -0.8 ±0.4 | 5.5 ±0.6 | 7.0 ±1.3 | 8.5 ±0.7 | -0.6 ±1.1 | 44.1 ±11.4 | 41.3 ±27.8 |
| Medium | HalfCheetah | 29.3 ±2.5 | 41.5 ±0.2 | 33.9 ±0.4 | 30.4 ±0.9 | 35.4 ±0.1 | 50.0 ±21.3 | 69.9 ±1.8 |
|  | Hopper | 31.5 ±5.4 | 38.4 ±12.9 | 88.5 ±0.1 | 74.8 ±15.9 | 37.6 ±4.6 | 85.1 ±3.8 | 92.8 ±1.6 |
|  | Walker2d | 43.1 ±9.7 | 46.9 ±1.7 | 70.4 ±0.3 | 61.5 ±1.0 | 59.6 ±0.4 | 44.1 ±11.4 | 62.8 ±5.8 |
| Medium-replay | HalfCheetah | 44.5 ±0.1 | 32.3 ±2.1 | 29.9 ±3.8 | 30.4 ±3.5 | 32.6 ±0.2 | 50.0 ±21.3 | 72.0 ±1.2 |
|  | Hopper | 24.7 ±1.5 | 35.2 ±2.0 | 29.6 ±2.8 | 39.4 ±14.7 | 65.9 ±14.0 | 85.1 ±3.8 | 92.3 ±3.2 |
|  | Walker2d | 21.1 ±0.7 | 14.5 ±1.3 | 20.0 ±3.2 | 38.0 ±4.3 | 46.4 ±3.7 | 44.1 ±11.4 | 81.1 ±8.1 |
| Medium-expert | HalfCheetah | 16.6 ±2.9 | 59.3 ±5.3 | 77.6 ±0.7 | 29.9 ±1.0 | 64.4 ±1.9 | 50.0 ±21.3 | 74.8 ±2.6 |
|  | Hopper | 68.7 ±6.8 | 38.7 ±11.8 | 88.8 ±4.4 | 50.9 ±9.5 | 88.0 ±5.5 | 85.1 ±3.8 | 93.6 ±0.8 |
|  | Walker2d | 38.2 ±9.8 | 58.8 ±11.4 | 50.6 ±8.5 | 54.3 ±2.8 | 79.2 ±3.4 | 44.1 ±11.4 | 75.0 ±3.5 |
| Total Mean |  | 29.2 ±3.4 | 33.7 ±4.2 | 48.0 ±4.9 | 40.3 ±6.2 | 44.3 ±3.6 | 59.7 ±12.2 | 75.8 ±5.1 |

(b) Fine-tuning performance on novel online tasks

Table 11: Performance comparison on the same and novel online tasks. The novel online tasks are generated by adjusting the gravity, noisy scales, and reward functions of the original MuJoCo environments. ∗ indicates the results reported by the paper. Note that FamO2O runs 1M fine-tuning steps, which is impractically expensive and results in an unfair comparison. We fine-tune all other methods for 100K steps, using the same hyper-parameter settings provided by the original papers. Scores for each dataset are normalized according to the D4RL benchmark (Fu et al., 2020): **Normalized Score** $= 100 \times \frac{\text{return} - \text{random return}}{\text{expert return} - \text{random return}}$.

| Task | HiP3 | FOWM* |
|------|------|-------|
| hopper-medium-v2 | **107.7** | 100.7 |
| hopper-medium-replay-v2 | **102.9** | 93.5 |
| mean | **105.3** | 97.1 |

(a) Same tasks with FOWM

| Task | HiP3 | TD-MPC2 |
|------|------|---------|
| halfcheetah | **70.1** | 37.3 |
| hopper | **94.1** | 3.1 |
| walker2d | **65.1** | 2.8 |
| mean | **76.4** | 14.4 |

(b) Novel tasks with TD-MPC2

Table 12: New baseline comparisons with model based reinforcement learning algorithms. * indicates results reported by the paper. FOWM (Feng et al.) is a recent model-based offline-to-online RL method that shares similar benchmarks with HiP3, As shown in Table 12a above, HiP3 consistently outperforms FOWM on the Hopper tasks, despite FOWM being fine-tuned with 500K environment steps during the online phase, while HiP3 only uses 100K steps. For TD-MPC2 (Hansen et al.), all results used a fixed online budget of 100K steps to fairly evaluate their online adaptation. Besides, we set the model size=5 as the TD-MPC2 paper recommended, and all other hyper-parameters were set as its default. As shown in Table 12b, we found out that HiP3 significantly outperforms TD-MPC2 when training with fewer steps.

| Task | HiP3 | CAL-QL | AWAC |
|------|------|--------|------|
| pen-human-v1 | 35.8 | 27.6 | **36.6** |
| pen-cloned-v1 | **54.6** | -3.2 | 47.3 |
| **mean** | **45.2** | 12.2 | 41.9 |

Table 13: Fine-tuning performance on novel online tasks based on Adroit-pen. Adroit-pen task (24-DoF Shadow Hand) is recognized as a more challenging manipulation task due to its high-dimensional action space and the need for precise multi-finger coordination. HiP3 achieves an average normalized score of 45.2, which is better than the result of CAL-QL and AWAC, demonstrating HiP3's effectiveness on diverse evaluation domains.

