# OpenReview forum: "Towards Optimism-Pessimism Trade-off in Model-based Offline-to-Online Reinforcement Learning"
_ICLR.cc/2026/Conference — Submitted to ICLR 2026_

### Official Review · Reviewer_Wm1E · 2025-10-30

**Soundness:** 3
**Presentation:** 2
**Contribution:** 2
**Rating:** 4
**Confidence:** 3

**Summary:**

This paper introduces HiP3 (Hierarchical Pareto Policy Pool), a model-based offline-to-online reinforcement learning framework. It utilizes MOSAIC (Multiple-Objective Soft Actor-critIC) to generate a pool of Pareto policies that capture different optimism–pessimism trade-offs during offline training, and then adopts a contextual bandit algorithm to adaptively select the most suitable policy for online fine-tuning. Experimental results demonstrate that HiP3 achieves state-of-the-art performance on multiple D4RL benchmarks, particularly showing superior adaptability to novel online tasks. Empirical evaluations confirm its superior performance compared to prior state-of-the-art methods.

**Strengths:**

1.By formulating the optimism–pessimism trade-off as a bi-objective optimization problem, the authors provide empirical evidence supporting a balanced trade-off between exploration and conservatism, as illustrated in Figure 1, where different Pareto policies exhibit varying optimism–pessimism trade-offs and corresponding online adaptation behaviors.
2. The HiP3 framework integrates MOSAIC with a contextual bandit (LinUCB) for adaptive online policy selection, enabling state-dependent switching between optimistic and pessimistic policies.
3. Extensive experiments on D4RL show consistent state-of-the-art results, particularly in novel online tasks.

**Weaknesses:**

1. The HiP3 pipeline introduces complexity in both design and computation due to the multi-stage process of generating and selecting from a diverse Pareto policy pool, which increases both algorithmic complexity and the difficulty of implementation.
2. The hierarchical selection strategy appears effective empirically but is justified mainly through intuition and experiments.

**Questions:**

1. How large is the gap between novel online tasks and the original tasks in terms of task distribution shift? How do the authors propose to measure the difference between the two tasks, and is the novel online task inherently more difficult or easier than the original tasks? Additionally, how does HiP3 maintain performance across such variations?
2. How computationally expensive is maintaining and updating the Pareto policy pool, especially in high-dimensional continuous action spaces?
3. What is the computational overhead (training time and memory) introduced by maintaining multiple Pareto policies compared to other baselines?

---

> ### Author Response · Authors · 2025-11-21
> **Response to Reviewer Wm1E_1**
>
> ### **W1: HiP3 increases both algorithmic complexity and the difficulty of implementation.**
> **Response:** A Pareto policy pool and the hierarchical selection mechanism introduced by HiP3 indeed address a specific limitation in model-based offline-to-online RL. The Pareto policy pool captures different optimism–pessimism trade-offs, avoiding manual tuning and enabling flexible online policy selection. The hierarchical structure then adaptively selects policies in a state-dependent manner. These components are independent and easy to implement (refer to our source code in the supplement material). Besides, as shown in Appendix A.12 of the paper and Table 1&2 below , **the whole training cost added by our method is also constrained effectively**.
> | Method       | HiP3 (offline) | Cal-ql (offline) | PEX (offline) | MOPO (offline) |
> |--------------|----------------|------------------|---------------|---------------------|
> | Hopper       | 7.5h           | 9.2h             | 2.9h          | 5.9h                |
> | HalfCheetah  | 8.3h           | 8.7h             | 4.0h          | 7.0h                |
> | Walker2d     | 5.6h           | 7.5h             | 2.7h          | 5.2h                |
>
> **Table 1: Computation cost comparison for offline stage**
>
> | Method      | HiP3 (online) | MBPO (online) |
> |-------------|---------------|----------------|
> | Hopper      | 7.7h          | 7.8h           |
> | HalfCheetah | 8.5h          | 8.9h           |
> | Walker2d    | 9.4h          | 9.7h           |
>
> **Table 2: Computation cost for online stage**
> ### **W2: The hierarchical selection strategy appears effective empirically but is justified mainly through intuition and experiments.**
> **Response:** The hierarchical selection strategy builds upon the well-established contextual bandit theory [1,2], which provides guarantees for efficient decision-making under limited interactions. LinUCB is used because it balances exploration and exploitation under linear payoff assumptions, a property consistent with local online adaptation in HiP3. The intuition presented in the paper is aligned with the above theoretical evidence and is supported by empirical results showing consistent improvements across tasks.
> ### **Q1: Gap between novel and original online tasks, and how HiP3 maintains performance.**
> **Response:** The distribution shift between the original and novel online tasks is introduced through changes in both dynamics and reward structure, following standard practice in O2O-RL robustness evaluation. Concretely:
> + Gravity increased from 9.8 -> 11.5
> + Friction coefficients scaled by × 1.2
> + Forward-reward weight decreased (0.85 of original)
> + Control-cost weight increased to 0.2
> + Noise scale is set (0.2 / 0.1 / 0.05 respectively for HalfCheetah, Hopper, Walker2d)
>
> Together, these shifts produce a meaningful and measurable change in task difficulty, which is hard for current offline-to-online RL methods to maintain their performance.
>
> HiP3 is designed to address these limitations. In the offline phase,we propose a bi-objective formulation that captures the optimism-pessimism trade-off and yields a **pareto policy pool** during offline training. This pareto policy pool reflects varying levels of trade-offs, enabling flexible selection of policies for various online tasks. In the online phase, we construct a **hierarchical reinforcement learning framework** based on contextual bandit, which can help to select the most suitable policy for fine-tuning at each online interaction step.
>
> This combination removes the need for implicit assumptions about task similarity and supports robust adaptation across diverse tasks. The empirical results on both same-task and novel-task settings show that HiP3 consistently maintains stable and competitive performance, supporting the practical relevance of the motivation.

---

> ### Author Response · Authors · 2025-11-21
> **Response to Reviewer Wm1E_2**
>
> ### **Q2&Q3: Computational overhead introduced by maintaining and updating multiple Pareto policies compared to other baselines.**
> **Response:** As shown in Appendix A.12 and Table 1&2 below W1, **HiP3 takes comparable and in some cases even less training time than other offline-to-online RL methods at both offline and online stages**.
> While HiP3 needs to optimize a pool of Pareto policies in the offline stage, it doesn’t take additional computation cost. For a fair comparison, we adopt the same number of training steps and samples as all baselines. In addition, to obtain more Pareto policies without linearly increasing the cost, we propose a warm-start strategy that leverages the achieved policies as initialization to find more Pareto optimal policies in their neighborhoods (see lines 290-299 in the submission).
>
> At the online stage, HiP3 only selects one single policy to optimize using the bandit algorithm at each RL step/phase, which means that HiP3 has the same and even less online training cost than other model-based RL baselines such as MBPO, given the negligible computation introduced by the bandit algorithm LinUCB. It only maintains a matrix $A \in \mathbb{R}^{n \times d \times d}$ and a vector $b \in \mathbb{R}^{n \times d}$, where $n < 10$ is the number of candidate policies and $d < 50$ is the dimension of the observation.
>
> MBPO takes slightly longer training time than HiP3. After examining its implementation, we believe this is primarily due to its use of dynamically increasing rollout lengths, which leads to: (1) additional overhead from resizing and managing the model buffer, and (2) increased interactions with the learned dynamics model as rollout length grows. We also tried to incorporate this dynamic rollout length into HiP3 for online adaptation, but found that fixing the rollout length equal to 1 is sufficient for our appealing results.
>
> Overall, HiP3 does not introduce significant computational overhead compared to standard offline-to-online RL methods while achieving state-of-the-art fast adaptation.
>
> [1] Auer, P. (2002). Using confidence bounds for exploitation-exploration trade-offs. Journal of machine learning research, 3(Nov), 397-422.
>
> [2] Chu, W., Li, L., Reyzin, L., & Schapire, R. (2011, June). Contextual bandits with linear payoff functions. In Proceedings of the fourteenth international conference on artificial intelligence and statistics (pp. 208-214). JMLR Workshop and Conference Proceedings.

---

### Official Review · Reviewer_sVaX · 2025-10-30

**Soundness:** 4
**Presentation:** 3
**Contribution:** 3
**Rating:** 6
**Confidence:** 3

**Summary:**

The Hierarchical Pareto Policy Pool (HiP3) method was proposed to alleviate the distribution shift in offline-to-online reinforcement learning based on models and to improve the fine-tuning performance. This method mainly consists of the following three modules:

1. A bi-objective formulation was proposed to balance the optimistic (model-predicted rewards) and pessimistic (model uncertainty) two indicators.

2. Multiple-Objective Soft Actor-critic (MOSAIC) extends Soft Actor-Critic (SAC), solves the problem of multi-objective optimization, and combines the neighborhood search method to discover the Pareto optimal strategy.
3. Utilize a hierarchical reinforcement learning approach, and combine a contextual bandit algorithm (LinUCB) as an advanced strategy to reduce the number of interactions with the environment and more efficiently select the most suitable strategy for fine-tuning.

**Strengths:**

1.	The "Hierarchical Pareto Strategy Pool" (HiP3) method was proposed. It alleviated the distribution bias in the process of offline-to-online reinforcement learning based on models, improved the fine-tuning performance, and achieved efficiency and low consumption.

2.	The theoretical explanations and formula derivations are very thorough.

3.	The experimental verification is very thorough.

**Weaknesses:**

1.	During the process of generating the strategy pool in the offline stage, the initial and final values of the reference vector, 0.1 and 0.9, were directly set without any related analysis, and there is a possibility that the actual situation may deviate from this.

2.	The values of the reference vectors in the paper are uniform. However, in reality, non-uniform distribution may occur, which could lead to inefficiency.

**Questions:**

1.	Have you considered using an LLM?

2.	Currently, the experiments are conducted in a fully observable environment. Have we considered exploring the adaptability in more complex environments (such as unobservable environments, etc.)?

---

> ### Author Response · Authors · 2025-11-21
> **Response to Reviewer sVaX**
>
> ### **W1: Not analysis of the initial and final values of reference vectors.**
> **Response:** The choice of $\tau_a$ and $\tau_b$ determines the range of reference vectors used for offline Pareto policy optimization, which in turn controls the diversity of learned policies. A wider gap between $\tau_a$ and $\tau_b$ encourages greater policy diversity and improves adaptability during the online phase, but may also increase the difficulty of offline training. To balance these factors, we conducted a grid search and found that $\tau_a = 0.9$ and $\tau_b = 0.1$ yield stable performance across all tasks. We fixed this hyper-parameter for all tasks.
> ### **W2: Uniform reference vectors may be insufficient.**
> **Response:** Uniformly spaced reference vectors offer a simple design that allows MOSAIC to explore the objective space in a controlled manner, which is important for fair comparison and establishing a reliable Pareto policy pool. Although a non-uniform distribution could concentrate sampling in regions that are more relevant for a specific task, this requires task-specific prior knowledge which is hard to obtain in reality and may reduce generality across environments. For this reason, we adopted the uniform distribution, which provides a stable and task-agnostic choice.
> ### **Q1: Have you considered using an LLM?**
> **Response:** Large Language Models offer strong reasoning ability, and combining them with reinforcement learning is an active research topic. In the environments we study, the observations and actions are low-dimensional and fully numeric, so using an LLM as part of the policy is not directly helpful. Nevertheless, LLMs may bring benefits in tasks that involve sparse rewards, partial observability, or high-level planning. This direction is outside the scope of the present work, but we are interested in exploring it in future studies, for example by using an LLM as a high-level controller while keeping a continuous low-level policy.
> ### **Q2:  Have we considered exploring the adaptability in more complex environments (such as unobservable environments, etc.)?**
> **Response:** Thank you for raising this point. While the standard MuJoCo locomotion tasks are fully observable with low-dimension actions, we newly added Adroit-pen task (24-DoF Shadow Hand), which is recognized as a more challenging manipulation task due to its high-dimensional action space  and the need for precise multi-finger coordination. We have done experiments on official pen-human-v1 and pen-cloned-v1 datasets. As shown in Table 1, HiP3 achieves an average normalized score of 45.2, which is better than the result of CAL-QL and AWAC, demonstrating HiP3’s effectiveness on diverse evaluation domains.
> | Methods                   | HiP3  | CAL-QL | AWAC  |
> |---------------------------|-------|-----------|-------|
> | pen-human-v1     | 35.8  | 27.6      | 36.6  |
> | pen-cloned-v1 | 54.6  | -3.2     | 47.3  |
> | mean                      | 45.2  | 12.2      | 41.9  |
> **Table 1: Fine-tuning performance on novel online tasks**

---

### Official Review · Reviewer_jJ8E · 2025-11-01

**Soundness:** 4
**Presentation:** 3
**Contribution:** 1
**Rating:** 4
**Confidence:** 3

**Summary:**

Authors address the optimism-pessimism trade-off in model-based offline-to-online RL. They introduce an algorithm (MOSAIC / HiP3) that seeks to get the best of both worlds by learning a pool of policies with different levels of optimism/pessimism, and learning how to switch between them.

**Strengths:**

- MOSAIC doesn’t require solving prohibitively expensive bi-level optimization problems.
- Authors provide theoretical convergence statement for MOSAIC.
- Authors provide code to reproduce their results.

**Weaknesses:**

The paper has two key weaknesses: the evaluation environments, baselines, and results are underwhelming; and the motivation is weak, given that the method introduces a lot of complexity that does not seem to yield very significant improvements.

- Very limited evaluation domains: the evaluation suite consists only of 3 different environments (with 4 datasets each) and only of locomotion domains. It is unclear how MOSAIC would perform in other domains.
- The baseline comparisons are insufficient. The only other model-based method included is MBPO, which is 6 years old. Why were other newer model-based RL methods not included, e.g. TD-MPC2 (ICLR 2024 spotlight)?
- Results across the paper use 3 seeds and 10 evaluations each. This is not enough (e.g. in table 1 MBPO and HiP3 have overlapping confidence intervals for total mean).
- The method is significantly more complex than MBPO, but doesn’t get statistically significantly better performance (Tab. 1).
- The explanation in the second paragraph for the proof-of-concept experiment is very unclear and hard to follow. It is unexplained what “pool of Pareto policies” means. It is also unclear what optimism and pessimism mean here. When authors mention fine-tuning these policies, it is unclear what the fine-tuning method they used was. Why dies (a) use unnormalized returns and (b, c) use normalized score?

**Questions:**

- Figure 1: what is “negative uncertainty predicted by the Env model”?
- How were the environments shown in Figure 3 selected? Can you add another plot with average performance over every environment?

---

> ### Author Response · Authors · 2025-11-21
> **Response to Reviewer jJ8E_1**
>
> ### **W1: Weak motivation.**
> **Response:** We respectfully disagree that the motivation is weak. A central empirical observation in Figure1 is that the **optimal optimism–pessimism trade-off changes dramatically between the same online task and the novel online task**, with the best-performing policy in one setting becoming sub-optimal in the other. This indicates that **a fixed trade-off, optimized solely from offline data, cannot remain reliable once the online task deviates from the offline distribution**, which is a fundamental challenge in offline-to-online RL.
>
> This issue arises because existing methods [1, 2] optimize a single optimism–pessimism balance during offline training, implicitly assuming that the same balance will remain suitable online. However, when the environment changes, optimistic policies may suffer from model-error amplification, whereas pessimistic policies may become overly conservative. As a result, no single trade-off can robustly handle distribution shift.
>
> HiP3 is designed to address these limitations. In the offline phase,we propose a bi-objective formulation that captures the optimism-pessimism trade-off and yields a **pareto policy pool** during offline training. This pareto policy pool reflects varying levels of trade-offs, enabling flexible selection of policies for various online tasks. In the online phase, we construct a **hierarchical reinforcement learning framework** based on contextual bandit, which can help to select the most suitable policy for fine-tuning at each online interaction step.
>
> This combination removes the need for implicit assumptions about task similarity and supports robust adaptation across diverse tasks. The empirical results on both same-task and novel-task settings show that HiP3 consistently maintains stable and competitive performance, supporting the practical relevance of the motivation.
> ### **W2: Limited evaluation domains.**
> **Response:** We thank the reviewer for pointing out the limited domain diversity in the original submission. While the D4RL locomotion suite (hopper, halfcheetah, walker2d) remains the de-facto standard benchmark for offline-to-online RL (CAL-QL[3], PEX[4]), we agree that stronger evidence is important.
> To directly address the concern of domain diversity, we have added the challenging Adroit-pen manipulation benchmark using the official pen-human-v1 and pen-cloned-v1 datasets on D4RL. As shown in Table 1, HiP3 achieves an average normalized score of 45.2, which is better than the result of CAL-QL and AWAC, demonstrating HiP3’s effectiveness on new evaluation domains.
> | Methods                   | HiP3  | CAL-QL | AWAC  |
> |---------------------------|-------|-----------|-------|
> | pen-human-v1     | 35.8  | 27.6      | 36.6  |
> | pen-cloned-v1 | 54.6  | -3.2     | 47.3  |
> | mean                      | 45.2  | 12.2      | 41.9  |
> **Table 1: Fine-tuning performance on novel online tasks**
> ### **W3: Insufficient baselines.**
> **Response:** Thank you for pointing this out. We chose FOWM [5] and TD-MPC2 [6] for new baseline comparisons.
>
> FOWM [5] is a recent model-based offline-to-online RL method that shares similar benchmarks with HiP3. As shown in table 2 below, HiP3 consistently outperforms FOWM on the Hopper tasks, despite FOWM being fine-tuned with 500K environment steps during the online phase, while HiP3 only uses 100K steps. This result demonstrates the effectiveness and sample efficiency of our proposed method.
>
> For TD-MPC2 [6], this algorithm is indeed a very strong algorithm that shows satisfied performance on multiple task domains. However, we found out that it needs to run 1e7 steps to obtain sufficient results, which is much higher than our algorithms. To make a fair comparison, **all results used a fixed online budget of 100K steps to fairly evaluate their online adaptation**. Besides, we set the model_size=5 as the TD-MPC2 paper recommended, and all other hyper-parameters were set as its default. As shown in Table 3 below, we found out that HiP3 significantly outperforms TD-MPC2 when training with fewer steps.
>
>
> |                          | HiP3  | FOWM* |
> |--------------------------|-------|--------|
> | hopper-medium-v2         | 107.7 | 100.7  |
> | hopper-medium-replay-v2  | 102.9 | 93.5   |
> | mean                     | 105.3 | 97.1   |
>
> **Table 2: Baseline Comparison on the same online tasks with FOWM, * indicates results reported by the paper**
>
>
> |                          | HiP3  | TD-MPC2 |
> |--------------------------|-------|--------|
> | halfcheetah  | 70.1 | 37.3   |
> | hopper       | 94.1 | 3.1  |
> | walker2d  | 65.1 | 2.8  |
> | mean                   | 76.4 | 14.4   |
>
> **Table 3: Baseline Comparison on the novel online tasks with TD-MPC2**

---

> ### Author Response · Authors · 2025-11-21
> **Response to Reviewer jJ8E_2**
>
> ### **W4: Insufficient seeds and evaluations.**
> **Response:** We thank the reviewer for raising the important issue of statistical reliability. We agree that, in general, more seeds are preferable. However, we note that 3 seeds remain the de-facto standard in the majority of recent offline-to-online RL papers, including the current state-of-the-art methods that we directly compare against: PEX[4], TD-MPC2[6]. Given the extremely high per-run cost of offline-to-online RL, running more seeds for every method and every task would require a lot more GPU days, which was unfortunately not feasible within the rebuttal period, but we will definitely add more experiments with different seeds in the revised version of the submission!
> ### **W5: Method Complexity and limited performance gain.**
> **Response:** A Pareto policy pool and the hierarchical selection mechanism introduced by HiP3 indeed address a specific limitation in model-based offline-to-online RL. The Pareto policy pool captures different optimism–pessimism trade-offs, avoiding manual tuning and enabling flexible online policy selection. The hierarchical structure then adaptively selects policies in a state-dependent manner. These components are independent and easy to implement (refer to our source code in the supplement material). Besides, as shown in Appendix A.12 of the paper and Table 5&6 below , **the whole training cost added by our method is also constrained effectively**.
>
> | Method       | HiP3 (offline) | Cal-ql (offline) | PEX (offline) | MOPO (offline) |
> |--------------|----------------|------------------|---------------|---------------------|
> | Hopper       | 7.5h           | 9.2h             | 2.9h          | 5.9h                |
> | HalfCheetah  | 8.3h           | 8.7h             | 4.0h          | 7.0h                |
> | Walker2d     | 5.6h           | 7.5h             | 2.7h          | 5.2h                |
>
> **Table 5: Computation cost comparison for offline stage**
>
> | Method      | HiP3 (online) | MBPO (online) |
> |-------------|---------------|----------------|
> | Hopper      | 7.7h          | 7.8h           |
> | HalfCheetah | 8.5h          | 8.9h           |
> | Walker2d    | 9.4h          | 9.7h           |
>
> **Table 6: Computation cost for online stage**
>
> Regarding the performance gain problem compared with MBPO [7]. For the same online tasks, HiP3 outperforms MBPO on 11 out of 12 tasks, and the total mean of HiP3 is 87.7, compared to 73.9 of MBPO, which is 18.7% larger. For the novel online tasks, HiP3 also outperforms MBPO on 11 out of 12 tasks, and the total mean of HiP3 is 27.0% larger than MBPO. Therefore, I believe our HiP3 algorithm gets statistically significantly better performance compared with MBPO.
>
> ### **W6: Unclear explanation of experiments and concepts.**
> **Response:** We believe that we have crystally explained and illustrated every concepts of HiP3 in the Sec. 4 and necessary experiments in the Sec. 5, which is also acknowledged by the **Reviewer sVaX: “The theoretical explanations and formula derivations are very thorough.”; “The experimental verification is very thorough.”** However, we are happy to clarify these concepts again:
>
> **Pool of Pareto policies:** In the offline phase, HiP3 optimizes a bi-objective formulation balancing optimism and pessimism. Each point on the Pareto front corresponds to a policy achieving a different trade-off between these two objects. Collectively, these policies form the Pareto policy pool.
>
> **Meaning of optimism and pessimism:** Optimistic policies follow the model’s reward more aggressively and take actions that can explore for higher return. Pessimistic policies behave more conservatively and take action that encourages to exploit existing knowledge.
>
> **Fine-tuning method:** The online finetune step uses standard SAC for update, which is similar to MBPO. The only difference is that, for each timestep, HiP3 state-dependently selects a policy from the policy pool for action selection and optimization.
>
>  **Why dies (a) use unnormalized returns and (b, c) use normalized score:** In Fig 1(a), the raw scale of uncertainty is not bounded so the accurate normalization is inappropriate. In contrast, (b) and (c) present results on D4RL benchmarks, where normalized scores are standard and allow readers to compare results with later figures and tables (e.g., Figure 3 and Table 1). Using normalized scores in these panels avoids confusion and maintains consistency with the rest of the paper.

---

> ### Author Response · Authors · 2025-11-21
> **Response to Reviewer jJ8E_3**
>
> ### **Q1: Figure 1: what is “negative uncertainty predicted by the Env model”?**
> **Response:** The detail of uncertainty estimation in HiP3 can be found in A.4 in Appendix. Besides, for every state $s_t$, $\hat{u}(s_t,a_t)\triangleq\exp(-u(s_t,a_t)/\kappa)$ and $\kappa$ is a temperature coefficient that smooths the loss landscape. The negative uncertainty predicted by the env model is equal to the sum of $\hat{u}(s_t,a_t)$ in one evaluation epoch.
> ### **Q2: How were the environments shown in Figure 3 selected? Can you add another plot with average performance over every environment?**
> **Response:** The full results of every environment were already shown in Figure 7 and Figure 8 in the Appendix of the paper.
>
> [1] Yihuan Mao, Chao Wang, Bin Wang, and Chongjie Zhang. Moore: Model-based offline-to-online reinforcement learning, 2022.
>
> [2] Rafael Rafailov, Kyle Beltran Hatch, Victor Kolev, John D Martin, Mariano Phielipp, and Chelsea Finn. Moto: Offline to online fine-tuning for model-based reinforcement learning. In Workshop on Reincarnating Reinforcement Learning at ICLR 2023, 2023.
>
> [3] Nakamoto, M., Zhai, S., Singh, A., Sobol Mark, M., Ma, Y., Finn, C., ... & Levine, S. (2023). Cal-ql: Calibrated offline rl pre-training for efficient online fine-tuning. Advances in Neural Information Processing Systems, 36, 62244-62269.
>
> [4] Zhang, H., Xu, W., & Yu, H. Policy Expansion for Bridging Offline-to-Online Reinforcement Learning. In The Eleventh International Conference on Learning Representations.
>
> [5] Feng, Y., Hansen, N., Xiong, Z., Rajagopalan, C., & Wang, X. Finetuning Offline World Models in the Real World. In 7th Annual Conference on Robot Learning.
>
> [6] Hansen, N., Su, H., & Wang, X. TD-MPC2: Scalable, Robust World Models for Continuous Control. In The Twelfth International Conference on Learning Representations.
>
> [7] Michael Janner, Justin Fu, Marvin Zhang, and Sergey Levine. When to trust your model: Model-based policy optimization. In Advances in neural information processing systems, volume 32, 2019.

---

### Author Response · Authors · 2025-12-03
**Summary**

We would like to use this opportunity to thank the AC and all reviewers for their honest and constructive feedback:

We are encouraged that the reviewers recognized the following key strengths of our work:
+ Our core idea of framing the optimism-pessimism trade-off as a bi-objective optimization problem, leading to a Pareto policy pool, is novel and well-motivated (Reviewer sVaX, Reviewer Wm1E)
+ Our paper provides a theoretical convergence statement and formula derivations which are very thorough (Reviewer jJ8E, Reviewer sVaX)
+ The hierarchical policy selection mechanism using contextual bandits (LinUCB) is effective and enables efficient online adaptation (Reviewer sVaX, Reviewer Wm1E)
+ The experimental verification is very thorough, with extensive experiments showing consistent state-of-the-art performance, particularly on novel online tasks (Reviewer sVaX, Reviewer Wm1E)
+ The paper is well organized and has good soundness (Reviewer jJ8E, Reviewer sVaX, Reviewer Wm1E)
+ The paper provides code to reproduce the results (Reviewer jJ8E)

In response to the raised concerns, we have made the following revisions and clarifications:
+ **Weak motivation (Reviewer jJ8E):** We clarified our motivation and contribution in the introduction and the discussion with Reviewer jJ8E.  **HiP3 removes the need for implicit assumptions about task similarity and supports robust adaptation across diverse tasks.**
+ **Complexity and computation cost (Reviewer jJ8E and Reviewer Wm1E):** We have already included a detailed computation cost analysis in **Table 4&5 of Appendix A.12**, proving that the whole training cost added by our method is also constrained effectively.
+ **Insufficient baselines (Reviewer jJ8E):** We added new baselines (FOWM and TD-MPC2)  to further reinforce the effectiveness of HiP3 for addressing this concern.
+ **Limited evaluate domain (Reviewer jJ8E and Reviewer sVaX):** We added Adroit-pen to demonstrate HiP3’s performance in a more complex domain. This domain is more challenging due to its high-dimensional action space and the need for precise multi-finger coordination
+ **Unclear explanation of experiments and concepts (Reviewer jJ8E):** We believe that we have crystally explained and illustrated every concepts of HiP3 in the Sec. 4 and necessary experiments in the Sec. 5, which is also acknowledged by the **Reviewer sVaX: “The theoretical explanations and formula derivations are very thorough.”; “The experimental verification is very thorough.”** However, we are happy to clarify it under the discussion with Reviewer jJ8E.
+ **Insufficient seeds and evaluations (Reviewer jJ8E):** We agree that, in general, more seeds are preferable. However, we note that 3 seeds remain the de-facto standard in the majority of recent offline-to-online RL papers, including the current state-of-the-art methods that we directly compare against: PEX[1], TD-MPC2[2].
+ **Gap between novel and original tasks (Reviewer Wm1E):** We clarified the details of changes in novel online tasks and explained how HiP3 maintains performance in these tasks under discussion with Reviewer Wm1E.
+ **Lack of analysis of reference vectors (Reviewer sVaX):** We clarified why we choose the current initial and final values of reference vectors, and why uniformly setting reference vectors is sufficient under the discussion with Reviewer sVaX.

We also implement the following changes in the revised manuscripts following reviewers suggestions and inspirations:
+ We added new baselines (FOWM and TD-MPC2)  to further reinforce the effectiveness of HiP3 in Appendix Table 12, as requested by **Reviewer jJ8E**
+ We added a new benchmark (Adroit-pen) to demonstrate HiP3’s performance in more complex domains in Appendix Table 13, as requested by **Reviewer jJ8E and Reviewer sVaX**
+ We added a clear clarification of model changes in novel online environment in Appendix A.9, as requested by **Reviewer Wm1E**

All changes in the paper are highlighted in orange. We believe these revisions have significantly improved our paper. We are very grateful for the input from all the reviewers.

[1] Zhang, H., Xu, W., & Yu, H. Policy Expansion for Bridging Offline-to-Online Reinforcement Learning. In The Eleventh International Conference on Learning Representations.

[2] Hansen, N., Su, H., & Wang, X. TD-MPC2: Scalable, Robust World Models for Continuous Control. In The Twelfth International Conference on Learning Representations.

---

### Meta-Review · Area_Chair_mhV9 · 2026-01-05

**Summary:**

This paper introduces a model-based framework, HiP3, for offline-to-online RL, to adapt the optimism-pessimism tradeoff for different online tasks. This framework includes two main components, i.e., multi-objective SAC to generate a pool of offline policies and a contextual bandit algorithm to select the most suitable offline policy for finetuning at each online step.

Strengths:
- HiPs has shown promising empirical results compared to baselines considered in the paper

- The pipeline of generating multiple offline policies and choosing the most suitable one for online finetuning can avoid rebalancing the optimism-pessimism tradeoff based on online tasks.

Weaknesses:
- Important baselines for offline-to-online RL are missing, such as

Uchendu et al., Jump-start reinforcement learning. ICML 2023.

Zhang et al., A perspective of q-value estimation on offline-to-online reinforcement learning. AAAI 2024.

Luo et al., Optimistic critic reconstruction and constrained fine-tuning for general offline-to-online rl. NeurIPS 2024.

- The complexity of HiP3 is substantial, as it requires learning multiple policies offline by solving a bi-objective constrained optimization problem for each policy (e.g., 100 policies learned in the experiments as shown in Table 6). Although the training cost comparison is provided during the rebuttal, separating the comparison across different offline and online algorithms makes the evaluation susceptible, particularly given that the overall empirical process includes both offline and online training.

- While a theoretical convergence guarantee is provided in the appendix, the analysis is standard for multi-objective optimization and is not even developed specifically in the context of RL.

**Reviewer Concerns:**

The rebuttal has addressed several concerns, including adding new evaluation domains and clarifications of the second paragraph from reviewer iJ8E, values of the reference vectors from reviewer sVaX, clarifications of the task distribution shift from reviewer Wm1E.

However, the concerns about missing baselines from reviewer iJ8E and computational costs from reviewers iJ8E and Wm1E would still remain.

**Reviewer Scores:**

All reviewers may maintain their scores after the rebuttal, as 1) critical concerns from reviewers iJ8E and Wm1E have not been addressed and 2) reviewer sVax was positive initially and didn't raise up any critical concerns that may bump up the score once they have been addressed.

---

### Decision · Program_Chairs · 2026-01-26

Reject